# *Bacteroides uniformis* Ameliorates Carbohydrate and Lipid Metabolism Disorders in Diabetic Mice by Regulating Bile Acid Metabolism via the Gut–Liver Axis

**DOI:** 10.3390/ph17081015

**Published:** 2024-08-01

**Authors:** Xue-Xue Zhu, Chen-Yang Zhao, Xin-Yu Meng, Xiao-Yi Yu, Lin-Chun Ma, Tian-Xiao Chen, Chang Chang, Xin-Yu Chen, Yuan Zhang, Bao Hou, Wei-Wei Cai, Bin Du, Zhi-Jun Han, Li-Ying Qiu, Hai-Jian Sun

**Affiliations:** 1Department of Basic Medicine, Wuxi School of Medicine, Jiangnan University, Wuxi 214122, China; xuexue.zhu@jiangnan.edu.cn (X.-X.Z.); 6222803024@stu.jiangnan.edu.cn (C.-Y.Z.); 6222809047@stu.jiangnan.edu.cn (X.-Y.M.); 6202805012@stu.jiangnan.edu.cn (X.-Y.Y.); 1282200206@stu.jiangnan.edu.cn (L.-C.M.); 1282200108@stu.jiangnan.edu.cn (T.-X.C.); 1282200102@stu.jiangnan.edu.cn (C.C.); 1282200207@stu.jiangnan.edu.cn (X.-Y.C.); 1282220302@stu.jiangnan.edu.cn (Y.Z.); 8201706060@jiangnan.edu.cn (B.H.); caiweiwei@jiangnan.edu.cn (W.-W.C.); 8166024037@jiangnan.edu.cn (B.D.); 2Department of Physiology, Eberhard-Karls-University of Tübingen, 72074 Tübingen, Germany; 3Department of Clinical Research Center, Jiangnan University Medical Center, Wuxi 214001, China; 9862023155@jiangnan.edu.cn; 4State Key Laboratory of Natural Medicines, China Pharmaceutical University, No. 24 Tongjia Lane, Nanjing 210009, China

**Keywords:** diabetes, gut microbiome, *Bacteroides uniformis*, bile acid, AMPK, TGR5

## Abstract

Background: Type 2 diabetes mellitus (T2DM) is a metabolic syndrome characterized by chronic inflammation, insulin resistance, and islet cell damage. The prevention of T2DM and its associated complications is an urgent public health issue that affects hundreds of millions of people globally. Numerous studies suggest that disturbances in gut metabolites are important driving forces for the pathogenesis of diabetes. However, the functions and mechanisms of action of most commensal bacteria in T2DM remain largely unknown. Methods: The quantification of bile acids (BAs) in fecal samples was performed using ultra-performance liquid chromatography–tandem mass spectrometer (UPLC-MS/MS). The anti-diabetic effects of *Bacteroides uniformis* (*B. uniformis*) and its metabolites cholic acid (CA) and chenodeoxycholic acid (CDCA) were assessed in T2DM mice induced by streptozocin (STZ) plus high-fat diet (HFD). Results: We found that the abundance of *B. uniformis* in the feces and the contents of CA and CDCA were significantly downregulated in T2DM mice. *B. uniformis* was diminished in diabetic individuals and this bacterium was sufficient to promote the production of BAs. Colonization of *B. uniformis* and intragastric gavage of CA and CDCA effectively improved the disorder of glucose and lipid metabolism in T2DM mice by inhibiting gluconeogenesis and lipolysis in the liver. CA and CDCA improved hepatic glucose and lipid metabolism by acting on the Takeda G protein-coupled receptor 5 (TGR5)/adenosine monophosphate-activated protein kinase (AMPK) signaling pathway since knockdown of TGR5 minimized the benefit of CA and CDCA. Furthermore, we screened a natural product—vaccarin (VAC)—that exhibited anti-diabetic effects by promoting the growth of *B. uniformis* in vitro and in vivo. Gut microbiota pre-depletion abolished the favorable effects of VAC in diabetic mice. Conclusions: These data suggest that supplementation of *B. uniformis* may be a promising avenue to ameliorate T2DM by linking the gut and liver.

## 1. Introduction

Type 2 diabetes mellitus (T2DM) is becoming a global health problem that affects hundreds of millions of individuals [1,2]. The diabetic epidemic poses an increasing burden on national health care systems worldwide due to the high prevalence of metabolic syndrome and obesity as well as changes in people’s lifestyle [3,4]. The number of T2DM patients is estimated to reach 592 million by 2035, and the effective management of the diabetic epidemic is highly imperative [5]. Hyperglycemia and hyperlipidemia are key features of T2DM, and these are closely linked with the occurrence and development of numerous complications, including heart failure, renal failure, neuropathy, blindness, amputations, and hepatic steatosis [6,7,8]. Among the target organs injured by diabetes, the liver is critically implicated in carbohydrate and lipid metabolism in which disturbed equilibrium in glucose and lipid metabolism leads to glucose and lipid deposition in hepatocytes [9,10,11]. Abnormal lipid metabolism leads to hepatic steatosis that further worsens the development of insulin resistance and inflammation in the liver [12]. In turn, hepatic insulin resistance and inflammation exacerbate the development of hepatic steatosis, forming a vicious cycle in the context of diabetes [13]. Accordingly, the appropriate manipulation of glucose and lipid metabolism in the liver system might be a feasible therapeutic strategy for the amelioration of insulin resistance, hyperglycemia, and hyperlipidemia in T2DM.

Recently, the interaction between the gut microbiota and host has received considerable interest since mounting evidence suggests that dysbiosis of the gut microbiota plays a pathogenic role in the progression of cardiovascular disorders, obesity, T2DM, and chronic inflammatory diseases [14,15]. A better understanding of the link between dysregulated gut microbiota and the host might result in novel therapeutic implications for T2DM. Malfunction of the gut microbiota leads to alterations in several metabolites, such as bile acids (BAs), branched-chain amino acids (BCAA), lipopolysaccharides (LPS), trimethylamine (TMA), short-chain fatty acids (SCFAs), and propionic acid imidazole (PAI) [16]. Diabetic mice transplanted with normal human fecal flora exhibit ameliorated glucose disorders by changing bacterial composition to produce more SCFAs [17,18]. On the contrary, transplantation of gut microbiota from T2DM patients into normal mice was found to disturb blood glucose by regulating the metabolism of BAs [19]. It has been speculated that the phyla Firmicutes, Bacteroidetes, Proteobacteria, Actinobacteria, and Fusobacteria account for 90% of the total microbiota in humans [20,21]. Probiotics including *Bifidobacterium* and *Lactobacillus*, and prebiotics such as oligofructose and inulin, are found to delay or reverse the development of T2DM [22,23]. A combination of *Bifidobacterium lactis* LMG P-28149 and *Lactobacillus rhamnosus* LMG S-28148 increases insulin sensitivity in high-fat diet (HFD)-induced obese mice [24]. It has been shown that *Bifidobacterium longum* and *Lactobacillus* upregulated GLP-1 and IL-10 expression in patients with obesity or T2DM, and suppressed lipid accumulation in adipocytes [25,26,27]. The mixture of *B. animalis subsp. lactis* LA804 and *Lactobacillus gasseri* LA806 attenuated body weight gain and limited obesity-related metabolic dysfunction and inflammation in a murine model of diet-induced obesity [28]. *Parabacteroides distasonis* (*P. distasonis*) attenuates insulin resistance by repairing the intestinal barrier and reducing inflammation in T2DM rats [29]. *P. distasonis* is also found to restrain body weight gain, hyperglycemia, and hepatic steatosis in ob/ob and high-fat diet (HFD)-fed mice [30]. These findings suggest that the gut microbiota and their metabolites may be significantly associated with the development and progression of T2DM. The liver synthesizes and conjugates primary BAs that are secreted into the intestine where they are deconjugated and transformed into secondary BAs by intestinal bacteria, promoting the activation of the nuclear farnesoid X receptor (FXR) [31]. Activation of FXR is a promising therapeutic approach for T2DM and non-alcoholic fatty liver disease (NAFLD) [31], indicating that the gut–liver axis functions as a new perspective for the management of T2DM. Recently, it has been established that the abundance of *Bacteroides uniformis* (*B. uniformis*) was present in lower proportions in T2DM patients [32]. A host of evidence showed that *B. uniformis* is mainly responsible for the synthesis of bile acids (BAs) [33,34,35]. It is highly probable that dysregulated *B. uniformis* and its metabolites including BAs are central players in the etiologies of diabetes.

Thus, the aim of the study is to investigate the role of the gut microbiota, specifically *B. uniformis* and its metabolites cholic acid (CA) and chenodeoxycholic acid (CDCA), in the pathogenesis and treatment of T2DM. The study seeks to explore the potential therapeutic benefits and mechanisms of supplementing these microbiota and their metabolites in T2DM. We also used a small molecule compound library to screen out a potential compound vaccarin (VAC) that had the most potential to promote the growth of *B. uniformis*, and then evaluated its subsequent impact on T2DM.

## 2. Results

### 2.1. Abnormalities in BA Metabolism in T2DM Mice

We measured the BA levels in mice using ultra-performance liquid chromatography–tandem mass spectrometer (UPLC-MS/MS). The total levels of BAs were elevated in serum and liver of T2DM mice (Figure 1a,b), with concomitant increases in total BA levels in the feces and the ratio of total unconjugated BAs to conjugated BAs, as well as the ratio of secondary BAs to primary BAs in T2DM mice (Figure 1c–e). Moreover, the levels of cholic acid (CA), ursodeoxycholic acid (UDCA), and chenodeoxycholic acid (CDCA) were significantly reduced in T2DM mice (Figure 1f). The T2DM mice exhibited higher levels of hyodeoxycholic acid (HDCA), deoxycholic acid (DCA), and lithocholic acid (LCA), but lower levels of Tauro β-muricholate acid (Tβ-MCA), Tauro ω-muricholate acid (Tω-MCA), Tauro ursodeoxycholic acid (TUDCA) and Tauro hyodeoxycholic acid (THDCA) (Figure 1g). Similarly, the T2DM mice showed decreased levels of CA and CDCA in the liver (Figure 1h,i), indicating that BA metabolism was disrupted in T2DM. 

### 2.2. B. uniformis Promotes CA and CDCA Production

To evaluate the effects of *B. uniformis* on metabolic disorders, we treated T2DM mice with vehicle (PBS) or *B. uniformis* by oral gavage daily for 4 consecutive weeks (Appendix A). Compared to the control mice, the T2DM group showed a significant reduction in body weight; this was not affected by administration of *B. uniformis* for a period of 4 weeks (Figure 2a,b). At the end of the experiments, fasting blood glucose (FBG) was lower in *B. uniformis*-treated mice when compared to the T2DM group (Figure 2c,d). Oral glucose tolerance test (OGTT) and insulin tolerance test (ITT) showed glucose intolerance and insulin resistance in T2DM mice, effects that were effectively improved by colonization of *B. uniformis* (Figure 2e–h). In keeping with this, *B. uniformis* efficiently improved carbohydrate and lipid metabolism disorders in T2DM mice by reducing the circulating levels of glucose (GLU), total cholesterol (TC), triglycerides (TG), and low-density lipoprotein (LDL) (Figure 2i–l). Interestingly, *B. uniformis* treatment had no effect on the serum levels of high-density lipoprotein (HDL) (Figure 2m).

T2DM mice treated with *B. uniformis* showed decreases in levels of serum alanine aminotransferase (ALT) and aspartate transaminase (AST), as well as downregulated liver TC and TG contents (Appendix A). By contrast, the *B. uniformis*-treated mice had higher levels of glycogen in the liver in comparison with the vehicle-treated T2DM mice (Appendix A). In line with the changes in glucose and lipid profiles, we found that *B. uniformis* greatly lowered the mRNA levels of glucose-6-phosphatase (G6pase), phosphoenolpyruvate carboxykinase (PEPCK), fatty acid synthase (FAS), and sterol regulatory element-binding protein 1 (SREBP1) in the liver of T2DM mice (Figure 2n). Correspondingly, *B. uniformis* administration alleviated liver lesions and increased the contents of liver glycogen in T2DM mice since the relative abundance of *B. uniformis* was partially restored after administration of *B. uniformis,* as indicated by H&E staining and PAS staining, respectively (Figure 2o,p). As stated above, our metabolomics analysis revealed that several conjugated BAs were increased and several unconjugated BAs were decreased in feces from T2DM mice (Figure 1f,g), thus indicating lower bile salt hydrolase (BSH) content. Consistently, the fecal contents of BSH were inhibited in T2DM mice; this was significantly restored by *B. uniformis* (Figure 2q). Likewise, supplementation of *B. uniformis* elevated the levels of CA and CDCA in the liver of T2DM mice (Figure 2r,s). Subsequently, we examined whether CA and CDCA were sufficient to improve the disturbance in glucose and lipid metabolism in T2DM mice. Similarly, CA and CDCA showed similar beneficial effects on glucose and lipid metabolism in T2DM mice (Figure 3 and Appendix A).

### 2.3. CA and CDCA as Well as B. uniformis Ameliorated Hepatic Insulin Resistance and Lipid Deposition by Acting on the TGR5/AMPK Signaling Pathway 

Our results showed that a combination of palmitic acid (PA) and oleic acid (OA) dose-dependently increased the production of glucose, TC and TG in HepG2 cells, effects that were maximal when HepG2 cells were treated with PA (1 mM) and OA (2 mM) (Appendix A). However, this combination using PA (0.5 or 1 mM) and OA (1 or 2 mM) was toxic to HepG2 cells (Appendix A). As such, we selected 0.2 mM of PA and 0.4 mM of OA to mimic hepatic insulin resistance and lipid deposition in vitro. As shown in Appendix A, PA and OA led to the production of GLU and TC in HepG2 cells, and these increases were largely abrogated by pretreatment with CA and CDCA, respectively. Interestingly, the combined application of CA and CDCA seems to confer better effects (Appendix A). Not surprisingly, pretreatment with CA and CDCA obviously attenuated the increases in the production of GLU, TC and TG in PA/OA-challenged cells (Figure 4a–c). When compared with control treatment, incubation with CA and CDCA could significantly promote the deposition of glycogen in hepatocytes exposed to PA and OA (Figure 4d). The role of CA and CDCA in scavenging lipid droplet deposition was further confirmed by Oil O red staining (Figure 4e). Similar to animal results of CA and CDCA, CA and CDCA were able to diminish the mRNA levels of G6Pase, PEPCK, FAS, and SREBP1 in PA/OA-challenged hepatocytes (Figure 4f). 

The obtained results showed that the mRNA levels of TGR5 were lower in hepatocytes upon PA/OA exposure, which were reversed by CA and CDCA (Figure 4g). As expected, CA and CDCA pretreatment significantly upregulated the protein levels of phosphorylated AMPK and TGR5 in hepatocytes induced by PA/OA (Figure 4h–j). The results were replicated in T2DM mice (Appendix A). In addition, we found *B. uniformis* treatment obviously elevated the protein levels of phosphorylated AMPK and TGR5 in the livers of T2DM mice (Appendix A). More importantly, silencing TGR5 overtly abolished the favorable effects of CA and CDCA on glucose and lipid metabolism in PA/OA-induced HepG2 cells (Appendix A). 

### 2.4. Vaccarin (VAC) Improved Carbohydrate and Lipid Metabolism Disorders by Promoting the Growth of B. uniformis in T2DM Mice 

We next screened the potential of prebiotics to stimulate the growth of *B. uniformis* since *B. uniformis* may be a promising probiotic in the treatment of T2DM. A large number of flavonoids are reported to effectively prevent or treat diabetes and its complications due to their diverse mechanisms of action, efficacy and safety. Thus, we collected 79 natural flavonoids to screen the most potential candidates that can promote the growth of *B. uniformis*. The results of large-scale bacterial culture and drug screening showed that VAC exhibited the strongest capability to facilitate the growth of *B. uniformis* (Appendix A). Then, we investigated whether VAC exerted anti-diabetic effects by regulating the growth of *B. uniformis* and the subsequent production of CA and CDCA. The overall animal experimental workflow is depicted schematically in Appendix A. The body weight of T2DM mice was significantly lower than that of the control group, while the body weight of mice in the VAC group was higher than that in the model group (Figure 5a,b). By contrast, the fasting blood glucose (FBG) level in T2DM mice was significantly reduced after VAC treatment (Figure 5c,d). VAC treatment showed efficacy in improving glucose intolerance in T2DM mice in an OGTT (Figure 5e,f). In addition, the VAC-treated mice exhibited a marginal effect on insulin resistance in the ITT compared with the T2DM mice (Figure 5g,h). Besides, VAC administration alleviated liver lesions and increased the contents of liver glycogen in T2DM mice, as evidenced by H&E staining and PAS staining, respectively (Figure 5o). In compliance with *B. uniformis* in vivo, VAC treatment apparently reduced hyperglycemia and hyperlipidemia (Figure 5i–n), as well as relieved hepatic dysfunction and lipid deposition in the livers of T2DM mice (Appendix A). Although VAC did not affect the abundance of total bacteria in T2DM mice, it did restore the abundance of *Bacteroides* and *B. uniformis* in the fecal tissues of T2DM mice (Figure 5p,q). 

### 2.5. VAC Was Less Effective in Improving the Status of T2DM Mice with Intestinal Microbiota Depletion

To confirm that VAC reduced blood glucose by regulating intestinal microbiota in mice, the mice were divided into T2DM, T2DM + combined antibiotics (ABX), T2DM + VAC, and T2DM + ABX + VAC groups (Figure 6a). RT-qPCR was used to detect the total intestinal flora of mice. It was found that the intestinal flora of mice was almost depleted after antibiotic intervention (Figure 6b). The FBG and the serum levels of GLU, TC, TG, LDL, AST, ALT, as well as liver index were significantly decreased in T2DM mice treated with VAC (Figure 6c–j), while the level of glycogen was higher in the T2DM + VAC group (Figure 6k). These changes were largely compromised when the gut microbiota was depleted. Unlike this, deletion of the gut microbiota had no effect on the ability of VAC to attenuate hepatic TC and TG contents (Figure 6l,m). However, the suppressive actions of VAC on the mRNA levels of G6Pase, PEPCK, FAS, and SREBP1 were strikingly prevented in T2DM mice when the gut microbiota was depleted (Figure 6n). 

## 3. Discussion

Insulin resistance and abnormal carbohydrate and lipid metabolism are the main independent risk factors for T2DM. Therefore, effective improvement of insulin sensitivity and glucolipid metabolism is the main way to prevent and treat the occurrence and development of T2DM [36]. As the current understanding of the complex interactions between the gut microbiota and the host continues to deepen, there is an urgent unmet need to clarify the exact functions of individual-specific bacteria in health and diseases, including T2DM. Actually, it is well accepted that T2DM has been linked to dysbiosis of gut microbiota. Hence, supplementation of probiotics or inhibition of pathogenic gut bacteria might be proposed as an alternative and complementary therapy to control T2DM. As Gram-negative, obligatory anaerobic rods, *Bacteroides* are abundantly expressed in human gut microbiota [37,38]. As a species of *Bacteroides*, *B. uniformis* is inversely associated with serum LDL-cholesterol levels [39]. *B. uniformis* CECT 7771 is documented to prevent obesity and metabolic disorders [40,41,42,43,44,45]. However, it is unclear whether supplementation with *B. uniformis* improves glucose and lipid metabolism disorders in T2DM mice. Our RT-PCR results showed that the contents of *B. uniformis* were diminished in fecal samples of T2DM mice, and administration of *B. uniformis* ameliorated HFD-induced hyperglycemia, insulin resistance, and hyperlipidemia. Thus, *B. uniformis* may be a highly effective commensal bacterium for improving insulin resistance and lipid overproduction in HFD- and STZ-induced mice. Interestingly, the administration of *B. uniformis* only partially restored normal carbohydrate and lipid metabolism in T2DM mice, suggesting that other bacterial species may also play a significant role. Our ongoing research aims to identify and characterize these additional bacterial taxa. The partial restoration highlights the complexity of the gut microbiota and its interactions with host metabolism. Future studies will involve exploring the synergistic effects of multiple beneficial bacteria to fully understand their collective impact on carbohydrate and lipid metabolism.

It is well known that the liver is the body’s key energy regulator and plays a crucial role in regulating glucose and lipid metabolism [46]. A number of studies have shown that the occurrence of T2DM is related to the disorder of glucose and lipid metabolism in the liver. The liver and gastrointestinal tract are closely linked in terms of metabolic activity and immune response, mainly due to their close anatomical and physiological relationship. Our results showed that the impaired glucose and lipid metabolism was attenuated in T2DM mice after administration of *B. uniformis*. In other words, *B. uniformis* was found to retard glycogenesis and lipolysis in the liver, thus lowering circulating glucose and lipids in T2DM. Notably, further studies are highly required to examine whether *B. uniformis* and its metabolites play an anti-diabetic role by regulating other tissues, such as fat, muscle, and pancreas. 

The metabolites derived from the gut microbiota, such as BAs, SCFAs, amino acid derivatives, and lipopolysaccharides, are critical signaling molecules linking the gut microbiota with the host [47,48]. The biosynthesized BAs in the liver are critically involved in the regulation of lipid absorption and metabolism [49]. *B. uniformis* is capable of regulating functional BA metabolism [50]. To explore the underlying mechanism of action of *B. uniformis* on T2DM, we investigated metabolites produced by *B. uniformis*. After oral administration of *B. uniformis*, the levels of CA and CDCA in the feces of T2DM mice were increased, confirming that *B. uniformis* possesses a strong ability to regulate BA metabolism. More importantly, CA and CDCA conferred similar anti-diabetic effects as *B. uniformis*. As a consequence, the changes in BA metabolism induced by *B. uniformis* play multiple roles in improving glucose and lipid metabolites in T2DM mice. However, the exact amounts of bile acids produced by *B. uniformis* in vivo were not directly measured. Future studies will aim to quantify these BA levels to provide a more accurate comparison. Notably, the current findings primarily demonstrate an association between *B. uniformis* administration and improvements in hepatic glucose and lipid metabolism. While our results suggest that *B. uniformis* may promote the formation of CA and CDCA, further experiments are needed to establish a direct causal relationship. These future studies will include quantitative measurement of bile acids produced by *B. uniformis*, usage of germ-free mice to isolate the effects of *B. uniformis*, and metabolomic analysis to track the specific pathways influenced by *B. uniformis* and bile acids.

TGR5, a G protein-coupled receptor for BAs, plays an important role in the control of glucose and lipid homeostasis [51,52]. Activation of TGR5 promotes the secretion of glucagon-like peptide-1 (GLP-1), thereby inducing insulin secretion and conserving glucose homeostasis [53]. Furthermore, TGR5 activation leads to lipolysis [54]. Activation of TGR5 accelerates energy metabolism by stimulating the activity of deiodinase 2 [55]. Inactivation of AMPK, a cAMP-dependent protein kinase, is linked to the development of metabolic syndrome, such as obesity, hypertension, and hyperglycemia [56]. Thus, we hypothesized that CA and CDCA alleviated glucose and lipid metabolism disorders in T2DM mice through the TGR5/AMPK signaling pathway. To validate this hypothesis, we examined the expression levels of TGR5 and phosphorylated AMPK. Our results demonstrated that CA and CDCA treatment restored the protein levels of TGR5 and phosphorylated AMPK in the liver of T2DM mice, which was supported by similar results from *B. uniformis*-treated mice. Additionally, silencing TGR5 significantly eliminated the effects of CA and CDCA in hepatocytes. These experiments highlighted the contribution of the TGR5/AMPK axis to CA/CDCA-mediated protection against T2DM. These results indicate that CA and CDCA produced by *B. uniformis* inhibit liver gluconeogenesis and lipolysis by acting on the TGR5/AMPK signaling pathway, thereby improving diabetes and its possible complications. 

Finally, a natural compound VAC showed favorable ability to prevent the development of hyperglycemia and hyperlipidemia in T2DM mice by facilitating the growth of *B. uniformis*, indicating that VAC may be used as a prebiotic agent to treat T2DM. VAC has been shown to protect the intestinal barrier and modulate the microbiota composition in T2DM mice using antibiotic treatment [57]. This protective role involves a broader range of microbial changes and is not limited to *B. uniformis*. While *B. uniformis* may contribute to the observed benefits, it is likely that other microbial taxa are also involved. Our current findings indicate that while *B. uniformis* is one of the beneficial bacteria influenced by VAC, the overall modulation of gut microbiota is a contributing factor to its therapeutic effects. To establish a more precise causal relationship, future studies will be performed to determine the specific effects of *B. uniformis* within the context of VAC treatment. Moreover, targeted depletion and supplementation experiments can be used to directly assess the role of *B. uniformis* in the presence of VAC. Metabolomic and microbiome analyses will be used to identify other key bacterial species influenced by VAC treatment in the setting of T2DM.

## 4. Materials and Methods

### 4.1. Reagents

VAC was purchased from Shanghai Shifeng Technology (Shanghai, China). Streptozocin (STZ) and D-glucose were acquired from Sigma (St. Louis, MO, USA). Insulin, cholic acid (CA) and chenodeoxycholic acid (CDCA) were bought from Solarbio (Beijing, China). *B. uniformis* was procured from China General Microbiological Culture Collection Center (Beijing, China). The nonspecific control small interfering RNA (siRNA) and TGR5 siRNA were constructed by Genomeditech Co. (Shanghai, China). Antibiotics (neomycin, ampicillin, neomycin sulfate and metronidazole) were purchased from Shanghai Sangon Biotech Co., Ltd. (Shanghai, China). The specific primers were synthesized by Shanghai Sangon Biotech Co., Ltd. (Shanghai, China). Horseradish peroxidase (HRP)-conjugated secondary antibodies were purchased from Proteintech (Wuhan, China). Antibody against β-actin (42 kDa) was obtained from Boster Biological Technology Co., Ltd. (Wuhan, China). Antibodies against p-AMPK (62 kDa) and TGR5 (35 kDa) were purchased from Beyotime (Shanghai, China) and Abcam (Cambridge, MA, USA), respectively. Antibody against total AMPK (62 kDa) was obtained from ABclonal (Wuhan, China). 

### 4.2. Animals

Six-week-old male C57BL/6J mice were purchased from the Model Animal Research Center of Nanjing University (Nanjing, China). All mice were placed in a temperature- and humidity-controlled environment with a 12 h light/dark cycle and free access to drinking water and food. After one week of adaptive feeding, T2DM mice were produced by feeding with a high-fat diet (21.8 kJ/g, 60% fat, D12492) for 4 weeks followed by a single intraperitoneal injection of streptozotocin (STZ, 120 mg/kg), while the control group was given the same volume of citric acid buffer (0.1 mM pH 4.5) [58]. One week after the injection of STZ, blood glucose levels were measured using the Accu-Chek II glucometer (Roche, Pleasanton, CA, USA), and fasting blood glucose more than 11.1 mM was regarded as the criterion for successful modeling [59]. Thereafter, T2DM mice were given *B. uniformis* (10^8^ CFUs), CA + CDCA (10 mg/kg) [60,61], or VAC (1 mg/kg) [62] by intragastric gavage under an HFD, and the control group was fed with a normal diet. The body weight and FBG were recorded at different times. 

### 4.3. Ethics Statement

All animal experiments were conducted in accordance with the guidance of the Institutional Animal Care and Use Committee at Jiangnan University, and obtained animal experiment approval on 15 September 2021. (Wuxi, China; Approval Number: No20210915c0600129).

### 4.4. Assessment of Biochemical Indicators

To evaluate glucose and lipid metabolism, serum samples were collected at the end of the experiments. The contents of glucose (GLU), free fatty acid (FFAS), total cholesterol (TC), triglyceride (TG), low-density lipoprotein (LDL), alanine aminotransferase (ALT), and aspartate transaminase (AST) in the serum of mice were determined using commercial kits according to the manufacturer’s protocols (Nanjing Jiancheng Bioengineering Institute, Nanjing, China). Also, the contents of TC, TG, and glycogen in the liver were detected in accordance with the instructions.

### 4.5. Oral Glucose Tolerance Tests (OGTT) and Insulin Tolerance Tests (ITT)

For OGTT measurement, all mice were fasted for 12 h, and D-glucose (2 g/kg) was orally gavaged. Also, after a 12 h fast, insulin (0.75 units/kg, i.p.) was injected intraperitoneally to test insulin sensitivity. FBG was measured with a blood glucose meter at 0, 15, 30, 60, 90, and 120 min after administration of glucose and injection of insulin.

### 4.6. Histological Staining Analysis

The collected liver specimens were fixed in 4% polyformaldehyde, dehydrated in ascending alcohol concentrations (70%, 95%, and 99% for 2 min each), cleared in xylene, and embedded in paraffin. Five-micrometer-thick sections were prepared from various animal groups, stained with hematoxylin and eosin (H&E), and examined under a light microscope [63]. The liver sections were subjected to Periodic Acid-Schiff (PAS) staining, which included the use of periodic acid solution and Schiff’s reagent (SolarBio, Beijing, China) to detect glycogen contents in the liver [64]. Microphotographs were taken using a microscope attachment camera (Nikon, Tokyo, Japan). 

### 4.7. Detection of Bacteria in Feces by Real-Time Fluorescence Quantitative Polymerase Chain Reaction (RT-PCR)

Fecal gDNA was extracted using a DNA Extraction Kit (cwbio, Taizhou, China). The extracted DNA was quantified with a Nanodrop spectrophotometer (Thermo Fisher Scientific, Waltham, MA, USA). The subsequent RT-PCR was performed in a total volume of 20 μL using 50 ng of gDNA, 200 nM forward and reverse primers, and qPCR SYBR Green Master Mix (YEASEN, Shanghai, China) on a LightCycler 480 II using Ultrasybr Mixture (Roche, USA). Relative gene expression was calculated by the 2^−ΔΔCt^ method using GAPDH as the internal reference normalized to the control group. The primers used are shown in Appendix A [65].

### 4.8. RT-PCR in the Liver Tissues and Cell Harvests 

Total mRNA in the liver tissues and cell harvests was extracted using a MolPure TRIeasy Plus Total RNA Kit (YEASEN, Shanghai, China) according to the manufacturer’s instructions. Total RNA was reverse transcribed using a HiScript III 1st Strand cDNA Synthesis Kit (+gDNA wiper) for qPCR according to the manufacturer’s instructions (Vazyme, Nanjing, China). Quantitative real-time PCR (qRT-PCR) was carried out using the Applied Biosystems QuantStudio 3 (ThermoFisher, USA) and Hieff qPCR SYBR Green Master Mix (Yeasen Biotech Co., Ltd., Sanghai, China). The mRNA levels of the genes were calculated by normalization to the levels of β-actin. The sequences used in this study are provided in Appendix A.

### 4.9. Western Blot

The Western blot protocol used was the same as we previously described [66]. The following antibodies were used: anti-p-AMPK antibody (AA393-1, Beyotime), anti-AMPK antibody (A1229, ABclonal), anti-TGR5 antibody (ab72608, Abcam), anti-β-actin (BM3873, Boster), HRP-linked anti-rabbit IgG (SA00001-2, Proteintech), and HRP-linked anti-mouse IgG (SA00001-1, Proteintech). Semi-quantitative analysis of each protein was performed using ImageJ software (version 1.5), and the band densities were normalized to the band density of β-actin.

### 4.10. Antibiotic Treatment

C57BL/6J male mice aged 6 weeks were fed a high-fat diet and combined with streptozotocin (STZ) to construct type 2 diabetes mice. They were then randomly divided into four groups: MOD group, MOD + ABX group, VAC group, and VAC + ABX group. After one week of adaptation, the mice were given a high-fat diet (21.8 kJ/g, 60% fat, D12492) for 4 weeks, followed by a single intraperitoneal injection of streptozotocin (STZ, 120 mg/kg), while the control group was given the same volume of citric acid buffer (0.1 mM pH 4.5). The mice were then subjected to gastric lavage with mixed antibiotics (1 g/L of ampicillin, 1 g/L of neomycin, 1 g/L of metronidazole, and 0.5 mg/L of vancomycin) or normal physiological saline. The Abx cocktail solution was freshly prepared every 2 d and administered continuously for one week [67]. After that, these mice were gavaged with VAC for 6 weeks. The weight and blood glucose of the mice were recorded every week. At the end of treatment, blood and main tissues were collected for further experiments.

### 4.11. Measurement of BAs

BA contents in the liver and feces samples were determined using UPLC-MS/MS (ACQUITY UPLC-XEVO TQ-S, Waters Corp., Milford, MA, USA) with an electrospray negative ionization source. The extraction method followed previous reports [68]. Briefly, liver and feces samples were homogenized in liquid nitrogen and suspended in ultrapure water, with 100 μL of each sample or serum used for analysis. After homogenization, the samples were centrifuged at 4 °C and 13,500 rpm for 20 min (Microfuge 20R, Beckman Coulter, Inc., Indianapolis, IN, USA). The supernatant was diluted 10 times with acetonitrile/methanol/water (*v*:*v*:*v* = 8:2:10). Samples were separated on a UPLC C18 column (2.1 × 100 mm, 1.7 μm; ACQUITY BEH, Waters) with a mobile phase of water (0.1% formic acid, A) and acetonitrile (B) at a flow rate of 0.3 mL/min. The elution gradient was 35% B (0–2 min), 35–90% B (2–10 min), and 90% B (10–13 min). BAs were detected using multiple reaction monitoring mode, and MS data were processed with MassLynx V4.1 software (Waters, Beverly, MA, USA).

### 4.12. Cell Culture 

Human liver carcinoma (HepG2) cells (French National Health Research Institute U553) were cultured in Dulbecco’s modified Eagle’s medium (DMEM, Hyclone, Logan, UT, USA) supplemented with 10% FBS (Lonsera, Shanghai, China) in a humidified 5% CO_2_ incubator at 37 °C. Oleic acid (OA) and palmitic acid (PA) were used to mimic insulin resistance and lipid overproduction in HepG2 cells. Cell viability was assessed using a cell counting kit-8 (CCK-8). The contents of TC, TG, and GLU in HepG2 cells were analyzed using specific biochemical assays. After treatment, HepG2 cells were washed with phosphate-buffered saline (PBS) and harvested. The harvested cells were lysed using RIPA buffer to release intracellular contents. Following the manufacturer’s instructions, the contents of TC, TG, and GLU were measured using commercial kits (Nanjing Jiancheng Bioengineering Institute, Nanjing, China). The results were normalized to the total protein content of the samples, which can be determined using a BCA protein assay.

### 4.13. Oil Red O Staining

HepG2 cells were cultured in a 12-well plate, pre-incubated with 50 μM CA and CDCA [69,70] for 30 min, and then treated with or without 0.2 mM of PA and 0.4 mM of OA. The cells were fixed with 4% paraformaldehyde for 30 min. The fixed cells were then stained with Oil Red O working solution for 15 min. After that, the working solution was discarded, and the cells were washed using 60% isopropanol for 30 s, followed by 2–3 washes with PBS buffer. The images were captured using a light microscope (Nikon, Japan).

### 4.14. siRNA Transfection

The control siRNA and TGR5 siRNA were transfected to HepG2 cells using lipofectamine^®^ 2000 according to the manufacturer’s instructions. After transfection for 6 h, the culture medium was replaced with fresh complete culture medium, and the cells were cultivated for another 24 h in the presence and absence of PA and OA.

### 4.15. Culture of B. uniformis

A total of 2 mL of culture medium was added to a sterile small culture tube, and the *B. uniformis* colonies on the blood plate were gently scraped off and seeded onto the culture tube under an anaerobic incubator. When obvious turbidity was observed in the culture tube, the microbial genome was extracted for bacterial identification. The DNA was isolated according to the instructions of the microbial genome extraction kit, and RT-PCR was subsequently conducted. The original bacterial solution (1 mL) was added to a 10 mL culture medium tube to passage *B. uniformis*. The number of *B. uniformis* was calculated using the colony plate counting method. One hundred μL of the bacterial solution was beaten onto a solid culture medium and evenly spread with a sterile coating rod. After anaerobic culture for 48 h at 37 °C, colony counting was carried out.

### 4.16. Screening of B. uniformis Growth Regulators

The cultivated *B. uniformis* was seeded in 96-well plates and incubated with 79 natural flavonoids (50 µM). After that, the OD600 value was measured after co-cultivation at different time points. After preliminary screening, the top ten potential candidates were added to the *B. uniformis*-loaded plates, and the OD600 value was measured time points of 0 h, 24 h, 48 h, and 72 h.

### 4.17. Measurement of BSH Contents

The contents of BSH in mice were measured using sandwich ELISA kits from Jiangsu Jingmei Biotechnology Co., Ltd. (Kunshan, China) as previously described [33]. Microtiter plates were coated with specific capture antibodies for mouse BSH. Monoclonal antibodies were applied to the wells, followed by the addition of the respective target proteins to form antibody-antigen-enzyme-labeled antibody complexes. After thorough washing, TMB substrate was added, which turned blue and then yellow upon catalysis by HRP enzyme and acid, respectively. The color intensity was directly proportional to the antibody levels in the samples. Protein content was determined by constructing a standard curve from the OD values measured at 450 nm using a microplate reader.

### 4.18. Statistical Analysis

The differences between the two groups were compared using the unpaired Student’s *t*-test, and comparisons among multiple groups were analyzed using ANOVA/Bonferroni-test. Data were expressed as mean ± standard deviation (SD). GraphPad Prism 9.0.1 was used for statistical analysis. *p* < 0.05 was considered statistically significant.

## 5. Conclusions

In the present study, we found for the first time that the abundance of *B. uniformis* in the fecal samples of T2DM mice was significantly decreased. Colonization of *B. uniformis* ameliorated hyperglycemia and hyperlipidemia, inhibited liver glycogenesis and lipolysis, and *B. uniformis*-derived metabolites CA and CDCA showed similar anti-diabetic effects. Mechanistically, CA and CDCA coordinately suppressed the overproduction of glucose and lipids in hepatocytes by activating the TGR5/AMPK signaling pathway (Figure 7). Eventually, high-throughput screening assays revealed that the natural flavonoid VAC strongly stimulated the growth of *B. uniformis*, with a concomitant improvement of T2DM. Thus, regulation of *B. uniformis* in the gut might be recognized as a key strategy to prevent or treat T2DM. Further studies are recommended to ascertain these findings in different T2DM mouse models, as well as investigate the safety and efficacy of *B. uniformis* and VAC in patients suffering from T2DM. The gut microbe *B*. *uniformis* and natural product VAC could represent a new strategy for the management of diabetes. *B*. *uniformis* was proven to be effective in treating T2DM by regulating the BA metabolism via the gut–liver axis, serving as a drug target for T2DM intervention. However, more research is warranted to verify the mechanism through which VAC modulated the gut microbiota and liver BA metabolism in the context of T2DM. In addition, it is deserving of investigation whether the administration of *B. uniformis* or CDCA and CA can restore intestinal dysbiosis in T2DM models.

## Figures and Tables

**Figure 1 pharmaceuticals-17-01015-f001:**
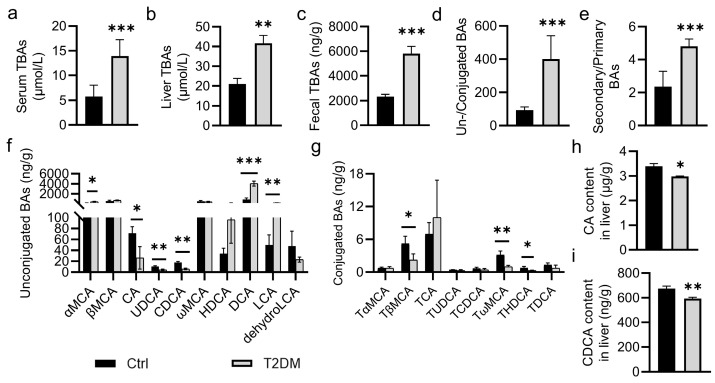
Abnormalities in bile acid metabolism in T2DM mice. (**a**) The level of serum total BAs (TBAs). (**b**) The level of liver TBAs. (**c**–**g**) Levels of BAs in fecal samples. (**h**,**i**) The levels of CA and CDCA. Data are represented as mean ± SD (n = 6). * *p* < 0.05, ** *p* < 0.01, *** *p* < 0.001 vs. Ctrl. Ctrl, control; T2DM, type 2 diabetes mellitus; TBAs, total bile acids; α-MCA, α-muricholate acid; β-MCA, β-muricholate acid; CA, cholic acid; UDCA, ursodeoxycholic acid; ω-MCA, ω-muricholate acid; HDCA, hyodeoxycholic acid; DCA, deoxycholic acid; LCA, lithocholic acid; dehydrol-CA; dehydrol-cholic acid; Tα-MCA, Tauro α-muricholate acid; Tβ-MCA, Tauro β-muricholate acid; TCA, taurocholic acid; TUDCA, Tauro ursodeoxycholic acid; TCDCA, Tauro chenodeoxycholic acid; Tω-MCA, Tauro ω-muricholate acid; THDCA, Tauro hyodeoxycholic acid; TDCA, Tauro deoxycholic acid.

**Figure 2 pharmaceuticals-17-01015-f002:**
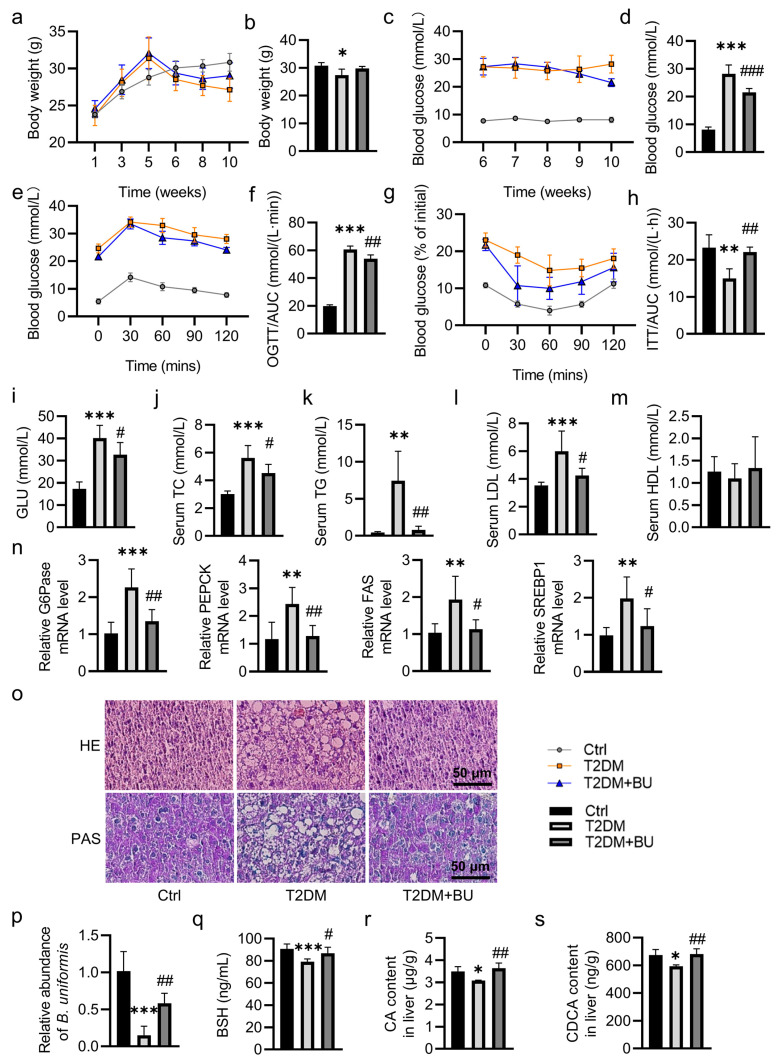
Effects of *B. uniformis* on glucose and lipid metabolism in T2DM mice. Male C57BL/6J mice were fed a high-fat diet for 4 weeks followed by a single intraperitoneal injection of STZ (120 mg/kg). In the sixth week, T2DM mice were given daily intragastric administration of vehicle (PBS) or *B. uniformis* for 4 consecutive weeks. (**a**) Body weight change curve and (**b**) the weight of mice at the end of the experiment. (**c**) FBG change curve and (**d**) FBG of mice at the end of the experiment. (**e**) Oral glucose tolerance test (OGTT) and (**f**) AUC of OGTT. (**g**) Insulin tolerance test (ITT) and (**h**) AUC of ITT. (**i**–**m**) The levels of serum GLU, TC, TG, LDL, and HDL. (**n**) The relative mRNA levels of G6Pase, PEPCK, FAS, and SREBP1. (**o**) Representative photographs of the liver with HE and PAS staining. (Scale bar = 50 μm). (**p**) Relative abundances of *B. uniformis* by qPCR. (**q**–**s**) The levels of BSH, CA and CDCA. Data are represented as mean ± SD (n = 6). * *p* < 0.05, ** *p* < 0.01, *** *p* < 0.001 vs. Ctrl; # *p* < 0.05, ## *p* < 0.01, ### *p* < 0.001 vs. T2DM. Ctrl, control; T2DM, type 2 diabetes mellitus; BU, *B. uniformis*; OGTT, oral glucose tolerance test; ITT, insulin tolerance test; GLU, glucose; TC, total cholesterol; TG, triglycerides; LDL, low-density lipoprotein; HDL, high-density lipoprotein; G6Pase, glucose-6-phosphatase; PEPCK, phosphoenolpyruvate carboxykinase; FAS, fatty acid synthase; SREBP1, sterol regulatory element binding protein 1; HE, hematoxylin-eosin; PAS, Periodic acid–Schiff; BSH, bile salt hydrolase; CA, cholic acid; CDCA, chenodeoxycholic acid.

**Figure 3 pharmaceuticals-17-01015-f003:**
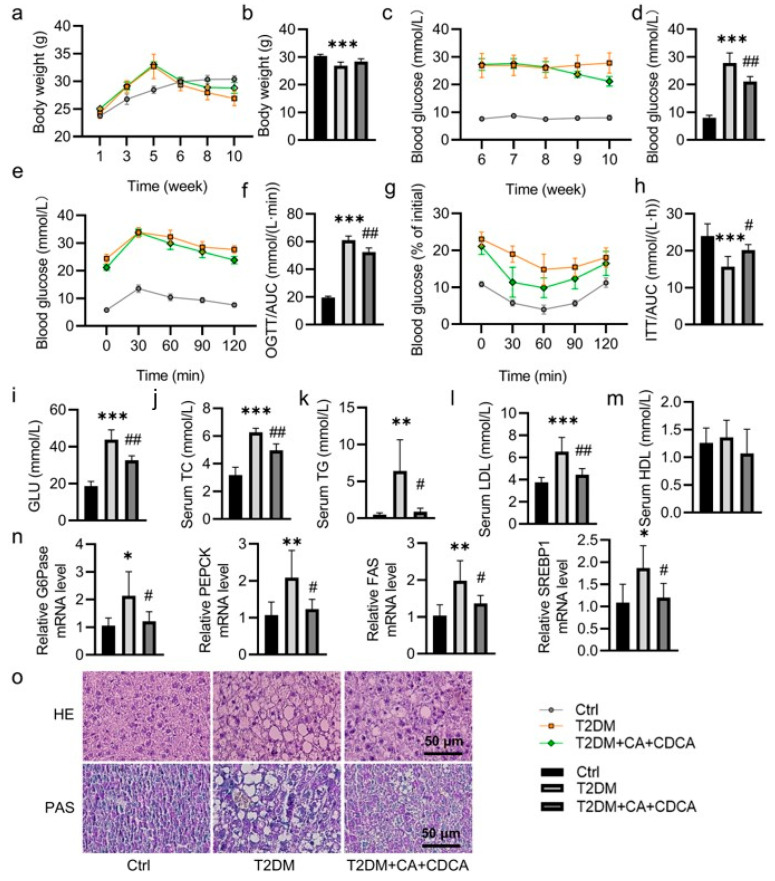
Effects of CA and CDCA on glucose and lipid metabolism in T2DM mice. Male C57BL/6J mice were fed a high-fat diet for 4 weeks followed by a single intraperitoneal injection of STZ (120 mg/kg). In the sixth week, T2DM mice were given daily intragastric administration of vehicle (PBS) or CA + CDCA (10 mg/kg) for 4 consecutive weeks. (**a**) Body weight change curve and (**b**) the weight of mice at the end of the experiment. (**c**) FBG change curve and (**d**) FBG of mice at the end of the experiment. (**e**) Oral glucose tolerance test (OGTT) and (**f**) AUC of OGTT. (**g**) Insulin tolerance test (ITT) and (**h**) AUC of ITT. (**i**–**m**) The levels of serum GLU, TC, TG, LDL, and HDL. (**n**) The relative mRNA levels of G6Pase, PEPCK, FAS, and SREBP1. (**o**) Representative photographs of the liver with HE and PAS staining. (Scale bar = 50 μm). Data are represented as mean ± SD (n = 6). * *p* < 0.05, ** *p* < 0.01, *** *p* < 0.001 vs. Ctrl; # *p* < 0.05, ## *p* < 0.01 vs. T2DM. Ctrl, control; T2DM, type 2 diabetes mellitus; OGTT, oral glucose tolerance test; ITT, insulin tolerance test; GLU, glucose; TC, total cholesterol; TG, triglycerides; LDL, low-density lipoprotein; HDL, high-density lipoprotein; G6Pase, glucose-6-phosphatase; PEPCK, phosphoenolpyruvate carboxykinase; FAS, fatty acid synthase; SREBP1, sterol regulatory element-binding protein 1; HE, hematoxylin-eosin; PAS, Periodic acid–Schiff.

**Figure 4 pharmaceuticals-17-01015-f004:**
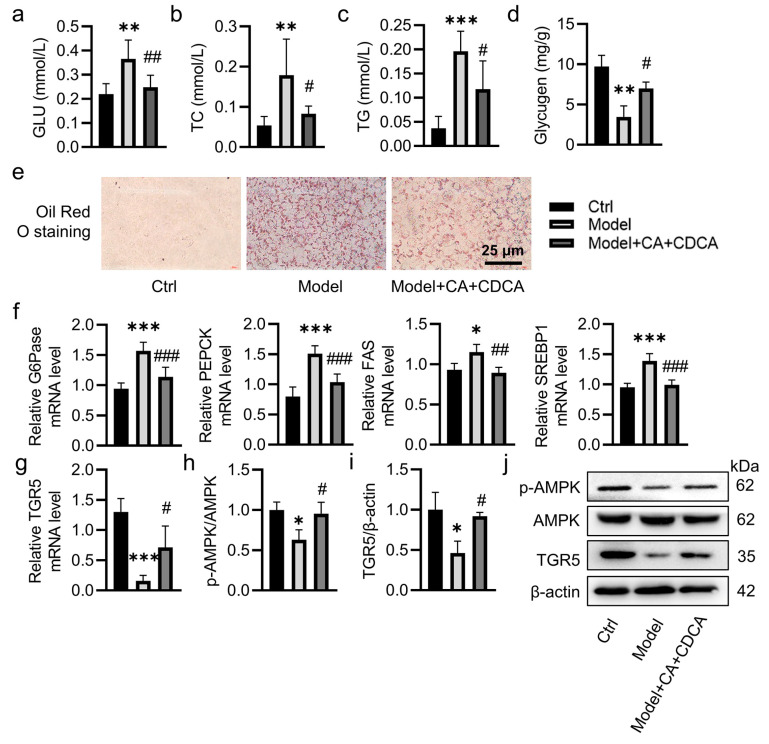
CA and CDCA ameliorated lipid deposition through the TGR5/AMPK signaling pathway. HepG2 cells were pre-incubated with 50 μM CA and CDCA for 30 min, and then treated without or with 0.2 mM of PA and 0.4 mM of OA for 24 h. (**a**–**d**) The content of GLU, TC, TG and glycogen. (**e**) Representative photographs of Oil Red O staining. (Scale bar = 25 μm). (**f**) The mRNA levels of G6Pase, PEPCK, FAS, and SREBP1. (**g**) The relative mRNA levels of TGR5. (**h**–**j**) Representative blots and quantification of p-AMPK/AMPK and TGR5. Data are represented as mean ± SD (n = 3–6). * *p* < 0.05, ** *p* < 0.01, *** *p* < 0.001 vs. Ctrl; # *p* < 0.05, ## *p* < 0.01, ### *p* < 0.001 vs. Model. Ctrl, control; CA, cholic acid; CDCA, chenodeoxycholic acid; GLU, glucose; TC, total cholesterol; TG, triglycerides; G6Pase, glucose-6-phosphatase; PEPCK, phosphoenolpyruvate carboxykinase; FAS, fatty acid synthase; SREBP1, sterol regulatory element-binding protein 1; TGR5, Takeda G protein-coupled receptor 5; AMPK, adenosine monophosphate-activated protein kinase; p-AMPK, phosphorylated adenosine monophosphate-actived protein kinase.

**Figure 5 pharmaceuticals-17-01015-f005:**
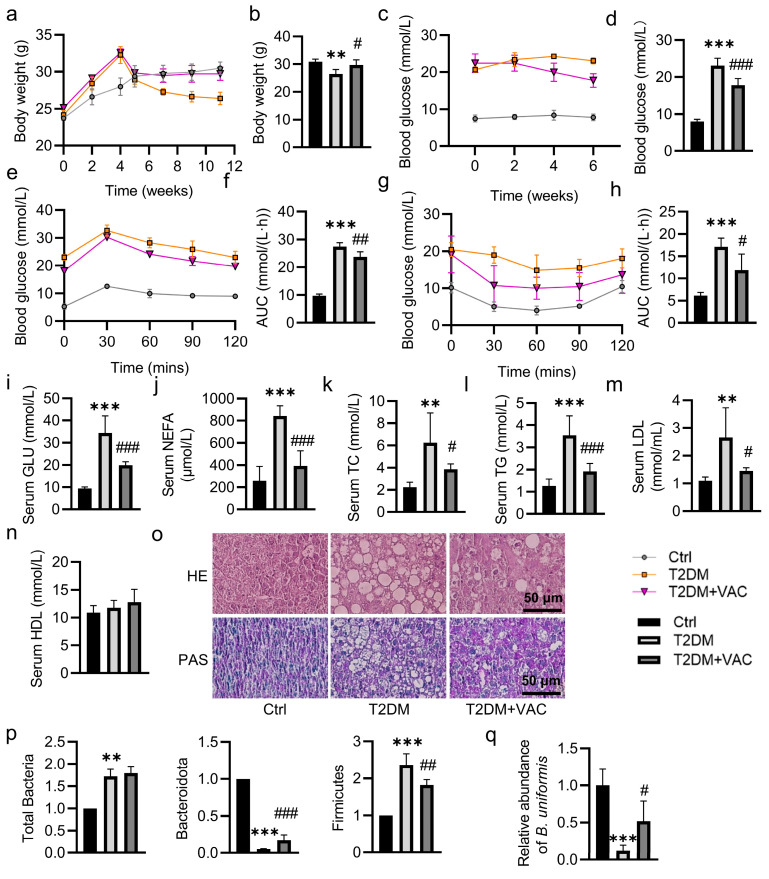
Effects of VAC on glucose and lipid metabolism in T2DM mice. Male C57BL/6J mice were fed a high-fat diet for 4 weeks followed by a single intraperitoneal injection of STZ (120 mg/kg). In the sixth week, T2DM mice were given daily intragastric administration of vehicle (PBS) or VAC (1 mg/kg) for 6 consecutive weeks. (**a**) Body weight change curve and (**b**) the weight of mice at the end of the experiment. (**c**) FBG change curve and (**d**) FBG of mice at the end of the experiment. (**e**) Oral glucose tolerance test (OGTT) and (**f**) AUC of OGTT. (**g**) Insulin tolerance test (ITT) and (**h**) AUC of ITT. (**i**–**n**) The levels of serum GLU, NEFA, TC, TG, LDL, and HDL. (**o**) Representative photographs of the liver with HE and PAS staining. (Scale bar = 50 μm). (**p**) The relative abundance of total bacteria, Bacteroides and Firmicutes by qPCR. (**q**) The relative abundance of *B. uniformis* by qPCR. Data are represented as mean ± SD (n = 6). ** *p* < 0.01, *** *p* < 0.001 vs. Ctrl; # *p* < 0.05, ## *p* < 0.01, ### *p* < 0.001 vs. T2DM. Ctrl, control; T2DM, type 2 diabetes mellitus; GLU, glucose; NEFA, non-esterified fatty acids; TC, total cholesterol; TG, triglycerides; LDL, low-density lipoprotein; HDL, high-density lipoprotein; HE, hematoxylin-eosin; PAS, Periodic acid–Schiff.

**Figure 6 pharmaceuticals-17-01015-f006:**
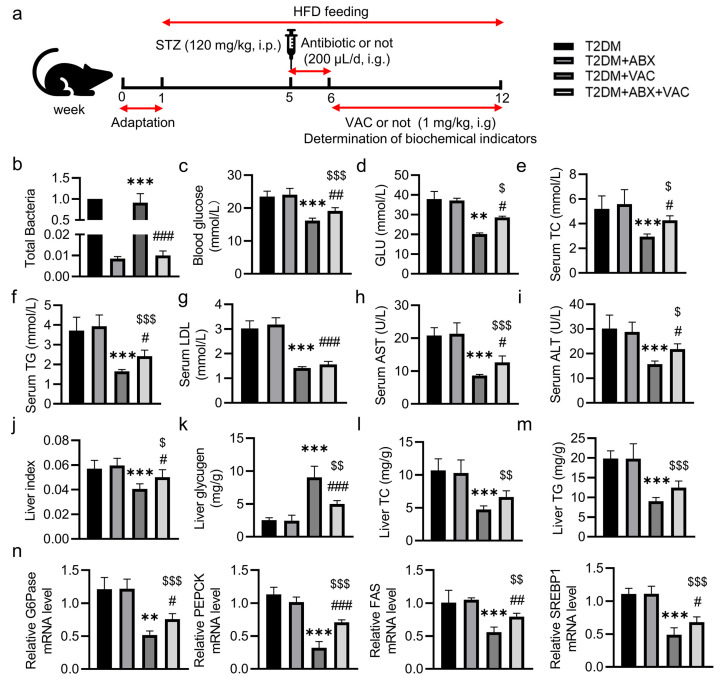
VAC was less effective in improving carbohydrate and lipid metabolism disorder in T2DM mice with intestinal microbiota deletion. Male C57BL/6J mice were fed a high-fat diet for 4 weeks followed by a single intraperitoneal injection of STZ (120 mg/kg). In the fifth week, the T2DM + ABX and T2DM + ABX + VAC groups were treated with ABX by oral gavage daily for one week. In the sixth week, T2DM mice were given daily intragastric administration of vehicle (PBS) or VAC (1 mg/kg) for 6 consecutive weeks. (**a**) Experimental protocol. (**b**) The relative abundance of total bacteria by qPCR. (**c**) FBG of mice at the end of the experiment. (**d**–**i**) The levels of serum GLU, TC, TG, LDL, AST and ALT. (**j**) The liver index. (**k**–**m**) The content of liver glycogen, TC and TG. (**n**) The relative mRNA levels of G6Pase, PEPCK, FAS, and SREBP1. Data are represented as mean ± SD (n = 6). ** *p* < 0.01, *** *p* < 0.001 vs. T2DM; # *p* < 0.05, ## *p* < 0.01, ### *p* < 0.001 vs. T2DM + VAC; $ *p* < 0.05, $$ *p* < 0.01, $$$ *p* < 0.001 vs. T2DM + ABX. T2DM, type 2 diabetes mellitus; STZ, streptozocin; HFD, high-fat diet; VAC, vaccarin; ABX, combined antibiotics; GLU, glucose; TC, total cholesterol; TG, triglycerides; LDL, low-density lipoprotein; AST, aspartate aminotransferase; ALT, alanine aminotransferase; G6Pase, glucose-6-phosphatase; PEPCK, phosphoenolpyruvate carboxykinase; FAS, fatty acid synthase; SREBP1, sterol regulatory element-binding protein 1.

**Figure 7 pharmaceuticals-17-01015-f007:**
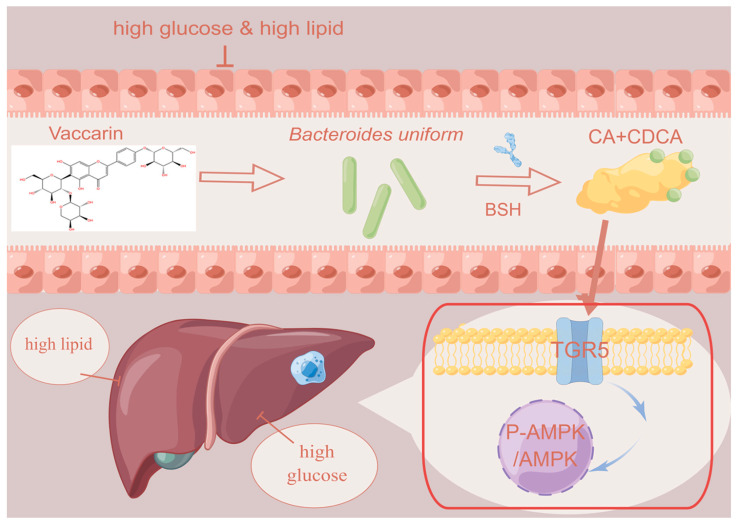
The mechanisms of VAC in ameliorating glucose and lipid metabolism disorders in T2DM.

## Data Availability

All data generated or analyzed during this study are included in this published article and Appendix A.

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
