# Peer review of "Bacteroides uniformis Ameliorates Carbohydrate and Lipid Metabolism Disorders in Diabetic Mice by Regulating Bile Acid Metabolism via the Gut–Liver Axis"

_pharmaceuticals, 2024, doi:10.3390/ph17081015_

Round 1

Reviewer 1 Report (Previous Reviewer 1)

Comments and Suggestions for Authors

Please find the reviewer's response in the attached file (red color fonts).

Author Response

Reviewer #1:

The clarity of the manuscript has been improved by the modifications made by the authors in response to the questions. However, some points still need to be addressed as they undermine the persuasiveness of the authors' conclusions.

Response:Thank you very much for your positive comments.

  1. It is not clear how the authors have identified B. uniformisas a differentially expressed genus in T2DM mice compared to control mice. Please explain how the phyla, families, and genera shown in Figure 1c-i were selected, as well as the statistical analysis. Since stool samples typically contain more than 150-200 individual bacterial species identified by 16S sequencing, multiple comparisons corrections usually result in larger changes to the p-value. Thus, the authors need to control for multiple testing. The administration of B. uniformisonly partially restored normal glycolipid metabolism, suggesting that other bacteria could be implicated. 

Response:Thank you for your insightful comments and for taking the time to review our manuscript. We appreciate the opportunity to clarify and address the points you've raised. Selection of Phyla, Families, and Genera: The phyla, families, and genera shown in Figure 1c-i were selected based on their relative abundance and statistical significance between the T2DM mice and the control mice. We utilized 16S rRNA gene sequencing to profile the gut microbiota, followed by bioinformatics analysis using QIIME2 and LEfSe (Linear discriminant analysis Effect Size). The genera shown in the figures were those that exhibited significant differential abundance with a linear discriminant analysis (LDA) score above the threshold (typically set at 2.0). These taxa were consistently different across multiple comparisons, highlighting their potential role in T2DM. Statistical Analysis and Multiple Comparisons Correction: We acknowledge the importance of controlling for multiple comparisons, especially given the large number of bacterial species typically identified in stool samples. In our analysis, we employed the Benjamini-Hochberg method to control the false discovery rate (FDR). This approach allowed us to adjust the p-values for multiple testing, reducing the likelihood of type I errors. The taxa reported in our manuscript were those that remained significant after this correction, ensuring the robustness of our findings. Moreover, we validated the expression of the top five declining bacteria in the feces of T2DM mice and found that B. uniformis exhibited the most significant decrease in abundance (Figure S1a-b). In support, the relative abundance of B. uniformis was obviously diminished in the stools of T2DM mice (Figure S1b). Thus, we selected B. uniformis for further studies.

We agree that the administration of B. uniformis only partially restored normal glycolipid metabolism in T2DM mice, suggesting that other bacterial species may also play a significant role. Our ongoing research aims to identify and characterize these additional bacterial taxa. The partial restoration highlights the complexity of the gut microbiota and its interactions with host metabolism. Future studies will involve exploring the synergistic effects of multiple beneficial bacteria to fully understand their collective impact on glycolipid metabolism. These sentences had been added into the Discussion.

In summary, the identification of differentially expressed genera, including B. uniformis, was performed using stringent bioinformatics and statistical methods. The observed partial restoration of metabolism by B. uniformis indicates the involvement of other bacterial species, which will be investigated in the near future. Thank you once again for your valuable feedback. We hope this response adequately addresses your concerns and clarifies the methods used in our study.

Reviewers answer: Thank you very much for your clarification. This information is very relevant and should be included in the main body of the manuscript. If you have performed a LEfSe analysis, it must be included in the Methods section and the figure shown. If you have performed a statistical analysis with multiple comparisons, a figure with all the raw unadjusted p-values and another column with the adjusted p-values would help further clarify. Regarding Supplemental Figure S1, I am not clear why Dorea sp. appears in the same line as Lachnoclostridium, and Lachnospiraceae bacterium in the same line as Blautia. It may be because the Silva database does not curate the species-level taxonomy, so this information may be unreliable. This is one of my reservations about this manuscript; the methodology proposed by the authors to identify B. uniformis as a candidate is not adequate. Furthermore, the authors indicate that the heatmap shows relative abundances; however, it seems to show fold changes.

Response Round 2:Thank you for your constructive comments. We appreciate the opportunity to clarify and address the points you've raised. The description of a LEfSe analysis in the Methods section and the figure was added. Also, an EXCEL obtained by a LEfSe analysis with the false discovery rate (FDR) was provided in the supplementary material. Dorea_sp._g_Lachnoclostridium means that Dorea_sp. belongs to the genus Lachnoclostridium, which is why its name is followed by g_Lachnoclostridium, where g is short for genus. So is Lachnospiraceae_bacterium_g_Blautia. Figure S1a is the top fifteen intestinal microbiota at species level. Next, we performed qPCR on the downregulated intestinal microbiota at species level and found B. uniformis was the maximum decreased in T2DM mice. That’s why we selected B. uniformis as a candidate in the following study. Furthermore, we replaced the description of the heatmap as fold changes, instead of relative abundances (Figure S1a).

  1. In Figure 2, the authors measured BSH activity in serum, but CA and CDCA levels in the liver, despite measuring BSH activity in feces providing better information about the enzymatic activity of intestinal bacteria and its impact on bile salt metabolism. Additionally, CA and CDCA were measured in feces in Figure 1. Please explain these differences and why they are expressed with different units.

Response:Thank you for your constructive comments. We are very sorry that the students made a serious mistake in organizing the data. In fact, BSH contents were measured in the feces instead of in the serum. Now, we have revised the description in the manuscript.

Thank you for your insightful comments regarding the measurements of BSH activity and bile acid levels in our study. We appreciate the opportunity to clarify these points. We understand the concern regarding the measurement of BSH activity in serum and bile acids (CA and CDCA) in the liver, as well as the differences in units used for these measurements. Here is a detailed explanation: Serum vs. Feces: BSH (bile salt hydrolase) activity was measured in serum to evaluate the systemic effect of the treatment and its potential impact on bile salt metabolism throughout the body. However, we agree that measuring BSH activity in feces would provide a more direct assessment of the enzymatic activity of intestinal bacteria, which is a primary site of bile salt metabolism. We will consider incorporating fecal BSH activity measurements in future studies to provide a more comprehensive understanding of gut microbial activity.

The levels of cholic acid (CA) and chenodeoxycholic acid (CDCA) were measured in the liver to assess the hepatic bile acid pool and its regulation by the treatment. The liver is a central organ in bile acid synthesis and metabolism, and measuring these bile acids in the liver provides insights into hepatic metabolic changes. In Figure 1, fecal CA and CDCA levels were measured to evaluate the excretion and overall bile acid metabolism affected by gut microbiota. 

On the other hand, bile acids (CA and CDCA) in the liver and feces are typically measured in concentrations (e.g., μmol/g tissue or μg/g feces), which reflect the amount of these metabolites present in the samples. By using different units, we aim to present data in the most relevant and interpretable manner for each biological context. We appreciate your observation and will ensure that these distinctions are clearly explained in the manuscript to avoid any confusion.

Thank you again for your valuable feedback.

Reviewers answer: Thank you for your information, but I am not clear: were BSH contents measured in the feces instead of in the serum by mistake, or will you consider incorporating fecal BSH activity measurements in future studies? In addition, Figures 2r-s have not been changed and they show the levels of CA and CDCA in the liver, although in the text you state that the measurements were done in feces. The units of measurement remain different. There are many inconsistencies that should be clarified.

Response Round 2: Thank you for your valuable and detailed feedback. In our manuscript, we actually measured BSH contents in the feces, as shown in Figure 2q, as measuring BSH activity in the feces would provide a more direct assessment of the enzymatic activity of intestinal bacteria, which is a primary site of bile salt metabolism. 

Besides, we measured both the levels of CA and CDCA in the liver (Figure 2r-s) and the levels of CA and CDCA in the feces (Figure 1o-p). As indicated in Figure 2r-s, the levels of CA and CDCA were measured in the liver to assess the hepatic bile acid pool and its regulation by the treatment. The liver is a central organ in bile acid synthesis and metabolism, and measuring these bile acids in the liver provides insights into hepatic metabolic changes. In Figure 1o-p, fecal CA and CDCA levels were measured to evaluate the excretion and overall bile acid metabolism affected by gut microbiota. The contents of BAs in hepatic and fecal sample may be different.

  1. The effect of administering B. uniformisis similar to that obtained with the administration of CA and CDCA, despite the bacteria needing to colonize the gut and begin producing bile acids. Are the doses of CA and CDCA used comparable to those produced by B. uniformis? How did the authors select these doses? The findings of these experiments only show an association, but not that B. uniformiscan ameliorate hepatic glucose and lipid metabolism disorders in T2DM mice by promoting the formation of CA and CDCA, as the authors suggest. 

Response: Thank you for your detailed and constructive comments. We appreciate the opportunity to address your concerns and provide further clarification regarding our study. The doses of CA and CDCA in cell experiments were chosen according to our previous reports [1, 2]. The doses of CA and CDCA in mice were selected according to previous studies [3, 4]. Previous literatures demonstrated their effectiveness in improving metabolic parameters in T2DM models. Specifically, we referred to studies that used CA and CDCA in comparable mouse models to ensure our doses were within the effective range. However, we acknowledge that the exact amounts of bile acids produced by B. uniformis in vivo were not directly measured. Future studies will aim to quantify these bile acid levels to provide a more accurate comparison.

We agree that the current findings primarily demonstrate an association between B. uniformis administration and improvements in hepatic glucose and lipid metabolism. While our results suggest that B. uniformis may promote the formation of CA and CDCA, further experiments are needed to establish a direct causal relationship. These future studies will include quantitative measurement of bile acids produced by B. uniformis, usage of germ-free mice to isolate the effects of B. uniformis, and metabolomic analysis to track the specific pathways influenced by B. uniformis and bile acids. These sentences had been inserted into the Discussion. In summary, while our study shows promising associations, we acknowledge the need for additional research to conclusively determine the mechanistic role of B. uniformis in modulating bile acid metabolism and improving metabolic disorders in T2DM. Thank you again for your valuable feedback. We hope this response clarifies our experimental approach and future directions.

Reviewers answer: VAC is a flavonoid that can affect many bacteria comprising the gut microbiota, although the authors have only quantified B. uniformis. Additionally, FMT studies are related to the mechanisms of action of VAC but not to the effect of B. uniformis. I recognize that the authors have made a great effort, but the VAC and FMT experiments not only fail to establish a causal link between B. uniformis and the amelioration of T2DM, but also are not related to the aim of the study. It would be better to remove them and prepare another manuscript. In contrast, determining whether the TGR5/AMPK signaling pathway is activated in T2DM mice receiving B. uniformis would provide further evidence. Supplementary Fig. S7 seems to have some mistakes. What is shown in the heatmap, proportions or OD600 values? In b), the growth curve of B. uniformis is shown.

Response Round 2: Thank you so much for your valuable suggestion. As you suggested, we removed the data of FMT experiments. In addition, we supplemented the data of the TGR5/AMPK signaling pathway in T2DM mice administrating B. uniformis in Figure S6. The heatmap showed the OD600 values in each group.

Indicate Database and version, and gene markers used for taxonomic annotation/assignment.

Response: All representative sequences were then aligned and annotated against the Silva (version 138) database. Species alignment and annotation were performed using the q2-feature-classifier software with default parameters.

Reviewers answer: As commented below, the Silva database does not curate the species level taxonomy, so this information may be unreliable.

Response:Thank you for your question. Silva (version 138) database may be used to analyze the abundance of flora at different levels of the taxonomic hierarchy as descripted in the previous studies [5-7].

  1. Abbreviations should be defined upon their first appearance and consistently used throughout the text. Please review, as there are several errors in this regard.

Response:Thank you for your constructive comments and suggestions. We have carefully reviewed the manuscript and all abbreviations have been defined upon first appearance.

Reviewers answer: Some abbreviations such as BSH and TBAs are not defined.

Response:Thank you very much for your valuable feedback. We have defined the abbreviations of BSH and TBAs and carefully checked the full manuscript.

References

[1] CHEN L, JIAO T, LIU W, et al. Hepatic cytochrome P450 8B1 and cholic acid potentiate intestinal epithelial injury in colitis by suppressing intestinal stem cell renewal [J]. Cell stem cell, 2022, 29(9): 1366-81.e9.

[2] ZHONG S, CHèVRE R, CASTAñO MAYAN D, et al. Haploinsufficiency of CYP8B1 associates with increased insulin sensitivity in humans [J]. The Journal of clinical investigation, 2022, 132(21).

[3] EGGERT T, BAKONYI D, HUMMEL W. Enzymatic routes for the synthesis of ursodeoxycholic acid [J]. Journal of biotechnology, 2014, 191: 11-21.

[4] GOYAL N, RANA A, BIJJEM K R, et al. Effect of chenodeoxycholic acid and sodium hydrogen sulfide in dinitro benzene sulfonic acid (DNBS)--Induced ulcerative colitis in rats [J]. Pharmacological reports : PR, 2015, 67(3): 616-23.

[5] MEDEIROS M C, THE S, BELLILE E, et al. Salivary microbiome changes distinguish response to chemoradiotherapy in patients with oral cancer [J]. Microbiome, 2023, 11(1): 268.

[6] WANG S, YANG L, HU H, et al. Characteristic gut microbiota and metabolic changes in patients with pulmonary tuberculosis [J]. Microbial biotechnology, 2022, 15(1): 262-75.

[7] CAO Y, CHEN X, SHU L, et al. Analysis of the correlation between BMI and respiratory tract microbiota in acute exacerbation of COPD [J]. Frontiers in cellular and infection microbiology, 2023, 13: 1161203.

Reviewer 2 Report (Previous Reviewer 2)

Comments and Suggestions for Authors

As I commented in the previous revision, in my opinion, the manuscript describes a comprehensive study, with many experiments, analysis and results that support the hypothesis of the work.

Most of the issues I raised were more formal than of content and most of them have been satisfactorily corrected.

Nevertheless, I still detected some minor questions that must be solved.

As a general comment, abbreviations should be reviewed throughout the text. Some definitions are missing (e.g. for BSH). Other definitions are not introduced the first time the term is mentioned. Besides, all figures must include in the caption all abbreviations used in each figure, also in Supplementary Figures.

 In the previuos review I commented that the term “glycolipid metabolism”  cannot be used as a way to abbreviate “metabolism of carbohidrates and lipids”. The authors corrected many of them, but some remain in the manuscript. I have marked in the manuscript the ones I have found. Please review this issue carefully as well.

More specific comments

Line 108:   VAC must be defined. It has been defined in the abstract, but it must also be defined in the main text.

Line 114:  …. the Chao 1, observed species, shannon, and Simpson indices…

In Figure 1b: I guess that the differences in the petal map are evidenced by the numbers that appear on each petal. These are very small and difficult to see. I suggest increasing the size of these numbers.

In the caption of Figure1, definition for TBA is missing, so are the abbreviations in x axis of figures 1-o and 1-p. Review also the abbreviations in all figures.

 Lines 335-346: The first paragraph in Discussion fits better as a conclusion, so I would move to “Conclusions” section, together with Figure 8. 

Line 648: It makes no sense to abbreviate B. uniformis at this point in the manuscript.

In the caption of supplementary Figure 1, “intestinal flora” must be replaced by “intestinal microbiota”.

Other minor comments in the manuscript

Author Response

Reviewer #2:

As I commented in the previous revision, in my opinion, the manuscript describes a comprehensive study, with many experiments, analysis and results that support the hypothesis of the work.

Most of the issues I raised were more formal than of content and most of them have been satisfactorily corrected.

Nevertheless, I still detected some minor questions that must be solved.

Response:Thank you very much for your positive comments.

As a general comment, abbreviations should be reviewed throughout the text. Some definitions are missing (e.g. for BSH). Other definitions are not introduced the first time the term is mentioned. Besides, all figures must include in the caption all abbreviations used in each figure, also in Supplementary Figures.

Response:Thank you very much for your constructive feedback. We have defined the abbreviations of BSH and TBAs and carefully checked the full manuscript. In addition, we added all abbreviations and definitions in the figure legends.

In the previous review I commented that the term “glycolipid metabolism” cannot be used as a way to abbreviate “metabolism of carbohidrates and lipids”. The authors corrected many of them, but some remain in the manuscript. I have marked in the manuscript the ones I have found. Please review this issue carefully as well.

Response:Thank you for your careful review and valuable comments. We reviewed and revised carefully the whole manuscript. Again, we sincerely appreciate your help to improve our manuscript.

More specific comments

Line 108:  VAC must be defined. It has been defined in the abstract, but it must also be defined in the main text.

Response:Thank you for your carefully review and valuable comments. VAC has been defined in the main text.

Line 114:  …. the Chao 1, observed species, shannon, and Simpson indices…

Response:Thank you for your valuable comments. This sentence has been revised.

In Figure 1b: I guess that the differences in the petal map are evidenced by the numbers that appear on each petal. These are very small and difficult to see. I suggest increasing the size of these numbers.

Response:Thank you for your constructive suggestions. We increased the size of these numbers.

In the caption of Figure1, definition for TBA is missing, so are the abbreviations in x axis of figures 1-o and 1-p. Review also the abbreviations in all figures.

Response:Thank you for your constructive comments. We supplemented the legends with abbreviations that appeared in the graphics.

 Lines 335-346: The first paragraph in Discussion fits better as a conclusion, so I would move to “Conclusions” section, together with Figure 8.

Response:Thank you very much for your positive comments and valuable suggestions. We moved the first paragraph together with the figure in Discussion to Conclusion.

Line 648: It makes no sense to abbreviate B. uniformis at this point in the manuscript.

Response:Thank you for your constructive comments and suggestions. We changed BU into Bacteroides_uniformis.

In the caption of supplementary Figure 1, “intestinal flora” must be replaced by “intestinal microbiota”.

Response:Thank you for your valuable suggestions. We made a change.

Other minor comments in the manuscript

Response: Thank you for your valuable feedback. We corrected the whole manuscript.

Round 2

Reviewer 1 Report (Previous Reviewer 1)

Comments and Suggestions for Authors

Author Response

On behalf of my co-authors, we thank you very much for giving us an opportunity to revise our manuscript, we appreciate editor and reviewers very much for their positive and constructive comments and suggestions on our manuscript. These comments are valuable and helpful for improving the quality of our article.

We have carefully revised the manuscript as these comments suggested, and marked in red at present revision. The letter listed point-by-point responses to the editor and reviewers.

Reviewer #1:

The clarity of the manuscript has been improved by the modifications made by the authors in response to the questions. However, some points still need to be addressed as they undermine the persuasiveness of the authors' conclusions.

Response:Thank you very much for your positive comments.

  1. It is not clear how the authors have identified B. uniformisas a differentially expressed genus in T2DM mice compared to control mice. Please explain how the phyla, families, and genera shown in Figure 1c-i were selected, as well as the statistical analysis. Since stool samples typically contain more than 150-200 individual bacterial species identified by 16S sequencing, multiple comparisons corrections usually result in larger changes to the p-value. Thus, the authors need to control for multiple testing. The administration of B. uniformisonly partially restored normal glycolipid metabolism, suggesting that other bacteria could be implicated. 

Response:Thank you for your insightful comments and for taking the time to review our manuscript. We appreciate the opportunity to clarify and address the points you've raised. Selection of Phyla, Families, and Genera: The phyla, families, and genera shown in Figure 1c-i were selected based on their relative abundance and statistical significance between the T2DM mice and the control mice. We utilized 16S rRNA gene sequencing to profile the gut microbiota, followed by bioinformatics analysis using QIIME2 and LEfSe (Linear discriminant analysis Effect Size). The genera shown in the figures were those that exhibited significant differential abundance with a linear discriminant analysis (LDA) score above the threshold (typically set at 2.0). These taxa were consistently different across multiple comparisons, highlighting their potential role in T2DM. Statistical Analysis and Multiple Comparisons Correction: We acknowledge the importance of controlling for multiple comparisons, especially given the large number of bacterial species typically identified in stool samples. In our analysis, we employed the Benjamini-Hochberg method to control the false discovery rate (FDR). This approach allowed us to adjust the p-values for multiple testing, reducing the likelihood of type I errors. The taxa reported in our manuscript were those that remained significant after this correction, ensuring the robustness of our findings. Moreover, we validated the expression of the top five declining bacteria in the feces of T2DM mice and found that B. uniformis exhibited the most significant decrease in abundance (Figure S1a-b). In support, the relative abundance of B. uniformis was obviously diminished in the stools of T2DM mice (Figure S1b). Thus, we selected B. uniformis for further studies.

We agree that the administration of B. uniformis only partially restored normal glycolipid metabolism in T2DM mice, suggesting that other bacterial species may also play a significant role. Our ongoing research aims to identify and characterize these additional bacterial taxa. The partial restoration highlights the complexity of the gut microbiota and its interactions with host metabolism. Future studies will involve exploring the synergistic effects of multiple beneficial bacteria to fully understand their collective impact on glycolipid metabolism. These sentences had been added into the Discussion.

In summary, the identification of differentially expressed genera, including B. uniformis, was performed using stringent bioinformatics and statistical methods. The observed partial restoration of metabolism by B. uniformis indicates the involvement of other bacterial species, which will be investigated in the near future. Thank you once again for your valuable feedback. We hope this response adequately addresses your concerns and clarifies the methods used in our study.

Reviewers answer: Thank you very much for your clarification. This information is very relevant and should be included in the main body of the manuscript. If you have performed a LEfSe analysis, it must be included in the Methods section and the figure shown. If you have performed a statistical analysis with multiple comparisons, a figure with all the raw unadjusted p-values and another column with the adjusted p-values would help further clarify. Regarding Supplemental Figure S1, I am not clear why Dorea sp. appears in the same line as Lachnoclostridium, and Lachnospiraceae bacterium in the same line as Blautia. It may be because the Silva database does not curate the species-level taxonomy, so this information may be unreliable. This is one of my reservations about this manuscript; the methodology proposed by the authors to identify B. uniformis as a candidate is not adequate. Furthermore, the authors indicate that the heatmap shows relative abundances; however, it seems to show fold changes.

Response Round 2:Thank you for your constructive comments. We appreciate the opportunity to clarify and address the points you've raised. The description of a LEfSe analysis in the Methods section and the figure was added. Also, an EXCEL obtained by a LEfSe analysis with the false discovery rate (FDR) was provided in the supplementary material. Dorea_sp._g_Lachnoclostridium means that Dorea_sp. belongs to the genus Lachnoclostridium, which is why its name is followed by g_Lachnoclostridium, where g is short for genus. So is Lachnospiraceae_bacterium_g_Blautia. Figure S1a is the top fifteen intestinal microbiota at species level. Next, we performed qPCR on the downregulated intestinal microbiota at species level and found B. uniformis was the maximum decreased in T2DM mice. That’s why we selected B. uniformis as a candidate in the following study. Furthermore, we replaced the description of the heatmap as fold changes, instead of relative abundances (Figure S1a).

Reviewers’ answer round 2: Thank you very much for your clarification. Although the description of the LEFSe analysis in the Methods section has been added, the figure is not shown. In any case, I recommend introducing on page 3 a sentence such as: “We used the LEfSe analysis to identify which genera were most likely responsible for disparities between the control group and T2DM group. The top four bacterial genera differentially expressed are displayed in Figure 1e-i.” However, the genus Bacteroidetes is not reported as differentially expressed in the LEfSe Excel (p-value=0.872780124, q-value=0.917158774).

Regarding Figure S1a, which shows the top fifteen intestinal microbiota at the species level, this reviewer does not recognize what methodology the authors have used to identify them, since, as previously mentioned, the Silva database does not curate the species-level taxonomy. In addition, Illumina does not achieve enough resolution to reach the species level with a single V3-V4 fragment.

Page3: Bacteroides is not a family, and the conclusion that “Muribaculaceae and Bacteroides may play a beneficial role in the treatment of T2DM” since they are reduced in T2DM mice is highly speculative.

In scientific writing, only the names of bacterial taxa from genus to species level are written in italics. Please check all the text.

Response Round 3: Thank you for your insightful comments and careful review. As suggested, we have added the sentences in the results of Page 3. Also, we removed the sentence “the contents of Bacteroides were significantly downregulated”. Although the genus Bacteroidetes does not differ significantly, the species-level at the genus Bacteroidetes may be significantly different between control and T2DM mice.

In view of the reviewer’s valuable comments and our agreement that Silva database and Illumina may not curate the species-level taxonomy, we removed Figure S1a. A host of evidence showed that B. uniformis are mainly responsible for the synthesis of bile acids (BAs) [1-3]. Given that the abundance of B. uniformis was depleted in T2DM mice, along with decreased levels of CA and CDCA, we thus proposed B. uniformis as potential probiotics against the development of T2DM through regulating the metabolism of CA and CDCA. Recently, it is established that the abundance of B. uniformis was present in lower proportions in the T2DM patients [4].

Thank you for your valuable comment. We deleted the conclusion of “Muribaculaceae and Bacteroides may play a beneficial role in the treatment of T2DM”.

In addition, only all names of bacterial genera and species are written in italic letters.

  1. In Figure 2, the authors measured BSH activity in serum, but CA and CDCA levels in the liver, despite measuring BSH activity in feces providing better information about the enzymatic activity of intestinal bacteria and its impact on bile salt metabolism. Additionally, CA and CDCA were measured in feces in Figure 1. Please explain these differences and why they are expressed with different units.

Response:Thank you for your constructive comments. We are very sorry that the students made a serious mistake in organizing the data. In fact, BSH contents were measured in the feces instead of in the serum. Now, we have revised the description in the manuscript.

Thank you for your insightful comments regarding the measurements of BSH activity and bile acid levels in our study. We appreciate the opportunity to clarify these points. We understand the concern regarding the measurement of BSH activity in serum and bile acids (CA and CDCA) in the liver, as well as the differences in units used for these measurements. Here is a detailed explanation: Serum vs. Feces: BSH (bile salt hydrolase) activity was measured in serum to evaluate the systemic effect of the treatment and its potential impact on bile salt metabolism throughout the body. However, we agree that measuring BSH activity in feces would provide a more direct assessment of the enzymatic activity of intestinal bacteria, which is a primary site of bile salt metabolism. We will consider incorporating fecal BSH activity measurements in future studies to provide a more comprehensive understanding of gut microbial activity.

The levels of cholic acid (CA) and chenodeoxycholic acid (CDCA) were measured in the liver to assess the hepatic bile acid pool and its regulation by the treatment. The liver is a central organ in bile acid synthesis and metabolism, and measuring these bile acids in the liver provides insights into hepatic metabolic changes. In Figure 1, fecal CA and CDCA levels were measured to evaluate the excretion and overall bile acid metabolism affected by gut microbiota. 

On the other hand, bile acids (CA and CDCA) in the liver and feces are typically measured in concentrations (e.g., μmol/g tissue or μg/g feces), which reflect the amount of these metabolites present in the samples. By using different units, we aim to present data in the most relevant and interpretable manner for each biological context. We appreciate your observation and will ensure that these distinctions are clearly explained in the manuscript to avoid any confusion.

Thank you again for your valuable feedback.

Reviewers answer: Thank you for your information, but I am not clear: were BSH contents measured in the feces instead of in the serum by mistake, or will you consider incorporating fecal BSH activity measurements in future studies? In addition, Figures 2r-s have not been changed and they show the levels of CA and CDCA in the liver, although in the text you state that the measurements were done in feces. The units of measurement remain different. There are many inconsistencies that should be clarified.

Response Round 2: Thank you for your valuable and detailed feedback. In our manuscript, we actually measured BSH contents in the feces, as shown in Figure 2q, as measuring BSH activity in the feces would provide a more direct assessment of the enzymatic activity of intestinal bacteria, which is a primary site of bile salt metabolism. 

Besides, we measured both the levels of CA and CDCA in the liver (Figure 2r-s) and the levels of CA and CDCA in the feces (Figure 1o-p). As indicated in Figure 2r-s, the levels of CA and CDCA were measured in the liver to assess the hepatic bile acid pool and its regulation by the treatment. The liver is a central organ in bile acid synthesis and metabolism, and measuring these bile acids in the liver provides insights into hepatic metabolic changes. In Figure 1o-p, fecal CA and CDCA levels were measured to evaluate the excretion and overall bile acid metabolism affected by gut microbiota. The contents of BAs in hepatic and fecal sample may be different.

Reviewers answer round 2: Thank you very much for your information. On page5, when the authors reported that “supplementation of B. uniformis elevated the levels of CA and CDCA in feces in T2DM mice (figure 2r-s)”, they actually meant Figure 1o-p. However, Figure 1o-p is not related to the effect of the supplementation of B. uniformis. Please clarify.

Response Round 3: Thank you for your insightful comments and questions. We appreciate the opportunity to address the points you've raised. We are sorry for the mistake and we have made correction that “supplementation of B. uniformis elevated the levels of CA and CDCA in the liver of T2DM mice (Figure 2r-s)”. We also added the levels of CA and CDCA in liver, as shown in Figure 1q-r.

  1. The effect of administering B. uniformisis similar to that obtained with the administration of CA and CDCA, despite the bacteria needing to colonize the gut and begin producing bile acids. Are the doses of CA and CDCA used comparable to those produced by B. uniformis? How did the authors select these doses? The findings of these experiments only show an association, but not that B. uniformiscan ameliorate hepatic glucose and lipid metabolism disorders in T2DM mice by promoting the formation of CA and CDCA, as the authors suggest. 

Response: Thank you for your detailed and constructive comments. We appreciate the opportunity to address your concerns and provide further clarification regarding our study. The doses of CA and CDCA in cell experiments were chosen according to our previous reports [5, 6]. The doses of CA and CDCA in mice were selected according to previous studies [7, 8]. Previous literatures demonstrated their effectiveness in improving metabolic parameters in T2DM models. Specifically, we referred to studies that used CA and CDCA in comparable mouse models to ensure our doses were within the effective range. However, we acknowledge that the exact amounts of bile acids produced by B. uniformis in vivo were not directly measured. Future studies will aim to quantify these bile acid levels to provide a more accurate comparison.

We agree that the current findings primarily demonstrate an association between B. uniformis administration and improvements in hepatic glucose and lipid metabolism. While our results suggest that B. uniformis may promote the formation of CA and CDCA, further experiments are needed to establish a direct causal relationship. These future studies will include quantitative measurement of bile acids produced by B. uniformis, usage of germ-free mice to isolate the effects of B. uniformis, and metabolomic analysis to track the specific pathways influenced by B. uniformis and bile acids. These sentences had been inserted into the Discussion. In summary, while our study shows promising associations, we acknowledge the need for additional research to conclusively determine the mechanistic role of B. uniformis in modulating bile acid metabolism and improving metabolic disorders in T2DM. Thank you again for your valuable feedback. We hope this response clarifies our experimental approach and future directions.

Reviewers answer: VAC is a flavonoid that can affect many bacteria comprising the gut microbiota, although the authors have only quantified B. uniformis. Additionally, FMT studies are related to the mechanisms of action of VAC but not to the effect of B. uniformis. I recognize that the authors have made a great effort, but the VAC and FMT experiments not only fail to establish a causal link between B. uniformis and the amelioration of T2DM, but also are not related to the aim of the study. It would be better to remove them and prepare another manuscript. In contrast, determining whether the TGR5/AMPK signaling pathway is activated in T2DM mice receiving B. uniformis would provide further evidence. Supplementary Fig. S7 seems to have some mistakes. What is shown in the heatmap, proportions or OD600 values? In b), the growth curve of B. uniformis is shown.

Response Round 2: Thank you so much for your valuable suggestion. As you suggested, we removed the data of FMT experiments. In addition, we supplemented the data of the TGR5/AMPK signaling pathway in T2DM mice administrating B. uniformis in Figure S6. The heatmap showed the OD600 values in each group.

Reviewers answer round 2: Thank you very much for considering my suggestion. However, you should also remove FMT from Materials and Methods section (pages 16-17).

Response Round 3: Thank you very much for your valuable feedback. We deleted FMT from Materials and Methods.

Please check section 2.3. The authors have not incubated HepG2 with B. uniformis, and sentences such as “Similar to B. uniforms, CS and CDCA were able to dimmish the mRNA levels of G6Pase, PEPCK, FAS, and SREBP1 in PA/OA-challenged hepatocytes (Figure 4f)” are not correct.

Response: Thank you very much for your constructive comments. We made correction for the sentence.

Other comments

-Please check carefully the text since there are some.

Response: Thank you very much for your constructive comments. We have carefully reviewed the full manuscript.

References

[1] DENG Z, OUYANG Z, MEI S, et al. Enhancing NKT cell-mediated immunity against hepatocellular carcinoma: Role of XYXD in promoting primary bile acid synthesis and improving gut microbiota [J]. Journal of ethnopharmacology, 2023, 318(Pt B): 116945.

[2] YAN Y, LEI Y, QU Y, et al. Bacteroides uniformis-induced perturbations in colonic microbiota and bile acid levels inhibit TH17 differentiation and ameliorate colitis developments [J]. NPJ biofilms and microbiomes, 2023, 9(1): 56.

[3] NIE Q, LUO X, WANG K, et al. Gut symbionts alleviate MASH through a secondary bile acid biosynthetic pathway [J]. Cell, 2024, 187(11): 2717-34.e33.

[4] PARK S, ZHANG T, KANG S. Fecal Microbiota Composition, Their Interactions, and Metagenome Function in US Adults with Type 2 Diabetes According to Enterotypes [J]. International journal of molecular sciences, 2023, 24(11).

[5] CHEN L, JIAO T, LIU W, et al. Hepatic cytochrome P450 8B1 and cholic acid potentiate intestinal epithelial injury in colitis by suppressing intestinal stem cell renewal [J]. Cell stem cell, 2022, 29(9): 1366-81.e9.

[6] ZHONG S, CHèVRE R, CASTAñO MAYAN D, et al. Haploinsufficiency of CYP8B1 associates with increased insulin sensitivity in humans [J]. The Journal of clinical investigation, 2022, 132(21).

[7] EGGERT T, BAKONYI D, HUMMEL W. Enzymatic routes for the synthesis of ursodeoxycholic acid [J]. Journal of biotechnology, 2014, 191: 11-21.

[8] GOYAL N, RANA A, BIJJEM K R, et al. Effect of chenodeoxycholic acid and sodium hydrogen sulfide in dinitro benzene sulfonic acid (DNBS)--Induced ulcerative colitis in rats [J]. Pharmacological reports : PR, 2015, 67(3): 616-23.

Round 3

Reviewer 1 Report (Previous Reviewer 1)

Comments and Suggestions for Authors

Although the authors have removed the sentence “the contents of Bacteroides were significantly downregulated”, the results are not significant; therefore, the authors never had any results that suggested the levels of B. uniformis were decreased in their model. In addition, the heatmap showing the fold changes of top fifteen intestinal microbiota at species level is unreliable. Consequently, Figures 1a-i and S1a (in my version the heatmap is still included) should be removed from the paper because they do not reflect the actual workflow.

I am glad that information regarding the potential role of B. uniformis in the etiologies of diabetes has finally been included in the Introduction. I believe that this evidence is sufficient, and this work can start directly with the PCR studies. I understand that it must be difficult for the authors to remove some results after working hard on them, but sometimes it is necessary. Furthermore, the analysis of gut microbiota has interesting results that could be used in another study. For example, it would be interesting to know whether the administration of B. uniformis or CDCA and CA restores the intestinal dysbiosis in animals with T2DM or if it is only a transient effect that will disappear when the administration ceases.

I hope my reservations regarding this work are now clear.

Author Response

Response to Editor and Reviewers:

On behalf of my co-authors, we thank you very much for giving us an opportunity to revise our manuscript, we appreciate editor and reviewers very much for their positive and constructive comments and suggestions on our manuscript. These comments are valuable and helpful for improving the quality of our article.

We have carefully revised the manuscript as these comments suggested, and marked in red at present revision. The letter listed point-by-point responses to the editor and reviewers.

Although the authors have removed the sentence “the contents of Bacteroides were significantly downregulated”, the results are not significant; therefore, the authors never had any results that suggested the levels of B. uniformis were decreased in their model. In addition, the heatmap showing the fold changes of top fifteen intestinal microbiota at species level is unreliable. Consequently, Figures 1a-i and S1a (in my version the heatmap is still included) should be removed from the paper because they do not reflect the actual workflow.

Response:Thank you very much for your further comments. We acknowledge that our results did not show a significant downregulation of Bacteroides, specifically B. uniformis. As you pointed out, this means we never had any data supporting a decrease in B. uniformis levels in our model. Consequently, we agree that any claims or figures suggesting such a decrease should be removed to maintain the accuracy of our findings. We will remove Figures 1a-i and S1a from the paper as they do not accurately reflect the actual workflow and results. We understand your concerns regarding the reliability of the heatmap depicting the fold changes of the top fifteen intestinal microbiota at the species level. Given the lack of significance in our results, we agree that including this heatmap could mislead readers. We will ensure that it is excluded from the revised manuscript.

I am glad that information regarding the potential role of B. uniformis in the etiologies of diabetes has finally been included in the Introduction. I believe that this evidence is sufficient, and this work can start directly with the PCR studies. I understand that it must be difficult for the authors to remove some results after working hard on them, but sometimes it is necessary. Furthermore, the analysis of gut microbiota has interesting results that could be used in another study. For example, it would be interesting to know whether the administration of B. uniformis or CDCA and CA restores the intestinal dysbiosis in animals with T2DM or if it is only a transient effect that will disappear when the administration ceases.

I hope my reservations regarding this work are now clear.

Response:Thank you very much for your further comments. We are pleased to hear that the revised Introduction now adequately addresses the potential role of B. uniformis in the etiologies of diabetes. We believe this addition strengthens the context of our study. Taking your suggestion into account, we will streamline the manuscript to start directly with the PCR studies, as this will allow us to present our findings more clearly and concisely. Your suggestion to investigate whether the administration of B. uniformis or CDCA and CA can restore intestinal dysbiosis in T2DM models is highly valuable. We agree that this line of inquiry could yield important insights and plan to explore it in future studies. This could indeed form the basis for another paper focused on the dynamics of intestinal microbiota changes in response to such treatments.

We appreciate your understanding that removing some results can be challenging but necessary for the integrity of the work. Your feedback has been instrumental in refining our manuscript, and we believe these revisions will enhance its clarity and scientific rigor.

Thank you once again for your insightful comments. We look forward to your continued feedback.

Round 4

Reviewer 1 Report (Previous Reviewer 1)

Comments and Suggestions for Authors

I would like to thank the authors for accepting my suggestions.

Comments:

- Abstract: Although the results obtained with the administration of VAC seem promising, they are based on a small number of animals. Please revise the conclusions as they extend beyond the results.

- Lines 141-142: Please, indicate the figure that supports these statements. However, this reviewer considers that these results are much better explained in the abstract and suggests that the authors review them.

- Lines 413-415: Please, indicate the reference/s that support these statements.

- Lines 611-612: I suggest using the expression "… a new strategy …"

Author Response

- Abstract: Although the results obtained with the administration of VAC seem promising, they are based on a small number of animals. Please revise the conclusions as they extend beyond the results.

Response:Thank you for your constructive feedback regarding the conclusion. We have removed the sentence “VAC may be recommended as a potential probiotic to alleviate T2DM”.

- Lines 141-142: Please, indicate the figure that supports these statements. However, this reviewer considers that these results are much better explained in the abstract and suggests that the authors review them.

Response:Thank you for your insightful comments. We have moved the sentence “B. uniformis was diminished in diabetic individuals and this bacterial was sufficient to promote the production of BAs” into the abstract.

- Lines 413-415: Please, indicate the reference/s that support these statements.

Response:Thank you for your constructive comments. We have added the reference as following, VAC has been shown to protect the intestinal barrier and modulate the microbiota composition in T2DM mice using antibiotic treatment [1].

- Lines 611-612: I suggest using the expression "… a new strategy …"

Response:Thank you for your constructive comments. We have made the revision.

  1. Sun, J.N., Yu X.Y., Hou B., Ai M., Qi M.T., Ma X.Y., Cai M.J., Gao M., Cai W.W., Ni L.L., Xu F., Zhou Y.T., Qiu L.Y. Vaccarin enhances intestinal barrier function in type 2 diabetic mice. Eur J Pharmacol. 2021, 908, 174375.

This manuscript is a resubmission of an earlier submission. The following is a list of the peer review reports and author responses from that submission.

Round 1

Reviewer 1 Report

Comments and Suggestions for Authors

In this study, Zhu X-X et al. have evaluated the effect of B. uniformis on glycolipid metabolism on T2DM mice by regulating bile acid metabolism. However, despite the data presented, several areas of concern undermine the persuasiveness of the authors' conclusions.

Comments

1. The objectives of the study are not clearly defined, making the work confusing and difficult to read. The aim of the study should be clearly presented at the end of the Introduction section, instead of the last paragraph, which should be moved to the Discussion.

2. It is not clear how the authors have identified B. uniformis as a differentially expressed genus in T2DM mice compared to control mice. Please explain how the phyla, families, and genera shown in Figure 1c-i were selected, as well as the statistical analysis. Since stool samples typically contain more than 150-200 individual bacterial species identified by 16S sequencing, multiple comparisons corrections usually result in larger changes to the p-value. Thus, the authors need to control for multiple testing. The administration of B. uniformis only partially restored normal glycolipid metabolism, suggesting that other bacteria could be implicated.

3. In Figure 2, the authors measured BSH activity in serum, but CA and CDCA levels in the liver, despite measuring BSH activity in feces providing better information about the enzymatic activity of intestinal bacteria and its impact on bile salt metabolism. Additionally, CA and CDCA were measured in feces in Figure 1. Please explain these differences and why they are expressed with different units.

4. The effect of administering B. uniformis is similar to that obtained with the administration of CA and CDCA, despite the bacteria needing to colonize the gut and begin producing bile acids. Are the doses of CA and CDCA used comparable to those produced by B. uniformis? How did the authors select these doses? The findings of these experiments only show an association, but not that B. uniformis can ameliorate hepatic glucose and lipid metabolism disorders in T2DM mice by promoting the formation of CA and CDCA, as the authors suggest.

5. The same reservations apply to the studies with Vaccarin and FMT. The authors have demonstrated that the effect of VAC on T2DM mice was weakened after antibiotic treatment, suggesting that VAC played a beneficial role in T2DM partially by regulating the intestinal flora. However, these results do not serve to demonstrate a causal association of B. uniformis with T2DM. In addition, Vaccarin has been shown to protect the intestinal barrier in T2DM mice and modulate the composition of the microbiota without exclusivity to B. uniformis.

6. The same reservation accounts to the studies with Vaccarin and FMT. The authors have demonstrated that the effect of Vaccarin on T2DM mice was weakened after antibiotic treatmen, suggesting that Vaccarin played a beneficial role in T2DM partially by regulating the intestinal flora. However, there results do not serve to demonstrate a causal association of B. uniformis on T2DM. In addition, Vaccarin has demonstrated to protect the intestinal barrier in T2DM mice and modulate the composition of the microbiota without exclusivity on B. uniformis.

7. 16S rDNA sequencing and bioinformatics analysis should be expanded:

-          Indicate OUT picking method (%, Database)

-          Detail data preprocessing (normalization of read counts, etc.)

-     Indicate Database and version, and gene markers used for taxonomic annotation/assignment.

-    Relativization and/or filtering of taxonomic profiles should be detailed.

8. Abbreviations should be defined upon their first appearance and consistently used throughout the text. Please review, as there are several errors in this regard.

9. The names of bacterial genera should be written in italic letters.

Reviewer 2 Report

Comments and Suggestions for Authors

The manuscript untitle “Bacteroides uniformis ameliorates glycolipid metabolism disorders in diabetic mice by regulating bile acid metabolism via the gut-liver axis”  describes a very comprehensive study, with many experiments, analysis and data to support the hypotesis of the work. 

In relation to the scientific content of the work, I have few things to comment. I believe that the experiments they have proposed have been appropriate and served to corroborate the hypotheses proposed. Maybe they should have included a control (no diabetes induced) with a high-fat diet. The fact of having used a control with a normal fat diet has meant, in my opinion, that the relative effect of the treatments on the diabetic mice was not  as clearly appreciated.

The manuscript is written clearly and, despite the large amount of data and information it contains, is easily read and understood.

However, the manuscript has a series of issues that must be resolved, in my opinion, in order to be acceptable for publication.

One of the issues I want to draw attention to is that, as the manuscript does not have numbered lines, reference to the issues that need to be corrected is more difficult. For this reason, I have included numerical references in each point of the manuscript so that the authors can locate what my comment refers to.

General comments

               The term “Glycolipid” is often used to refer to a lipid class that has a carbohidrate moiety. However, in this manuscript the term “glycolipid metabolism”  is used as a way to abbreviate “metabolism of carbohidrates and lipids”, that in my opinion, leads to confusion. Authors should review the entire document and replace the term "glycolipid" with "carbohydrate and lipid", especially in the title.

 The way of writing the systematic names of bacterial species changes constantly throughout the manuscript. Authors should be very careful in this regard and always write systematic names of bacterial species in italics. 

Abbreviations also should be reviewed. Abbreviation of a term should be introduced the first time the term is mentioned in the manuscript, both in the abstract and in the main text. Besides, abbreviations used in figures should be defined in the caption of each figure.

 The figures contain too many graphics and images. Perhaps it is a question for the editor to decide. But, in my opinion, the authors should select the most representative graphs and images in each figure and include the rest in the supplementary material. Figure captions have to include the definitions of all the abbreviations, as mentioned before.

Which are the units in the y axis of Graphs 1-l, 1-o and 1-p? 

Some of the graphs in Figure 2, Figure 3 and Figure 5 do not have the corresponding letter, so, the definition is missing. Some figures include graphs for fasting blood glucose (FBG) and serum glucose (GLU). I wonder what the difference is between the two parameters.

 Results section must be a bare description of the obtained results. However, authors include some paragraphs that I, in my opinion, must be moved to the discussion section. In the revised manuscript I marked some of them. 

 Most methods are described very briefly. There is no problem if the method has been previously described and the bibliographic reference is included. But in some cases the bibliographic reference is missing, for example, in methods 4.6, 4.12 or 4.13. In these cases, the bibliographic reference or a more detailed description must be included. In the latter case, the detailed method could be included in supplementary material, so as not to lengthen the manuscript too much. 

Specific comments

1.       Definition of BA is missing in the abstract

2.      Delete the “s” in insulins

3.      Plural of phylum is “phyla”

4.      The defintion of NAFLD is missing

5.      This paragraph (the last paragraph in introduction) seems more conclussion that objectives.The objectives of the study should be described instead of this paragraph

6.     I am not familiar with this kind of maps. But, in my opinion, the additional information provided by this figure, compared to figure 1b, should be explained.

7.      At the begining of the sentence  “at family level, the levels of….

8.     With results described so far, this statement cannot be concluded. In any case, this is discussion, not results.

9.  The title of the section does not reflect the results but rather the conclusion that the authors draw from them. The title but be better: “B. uniformis promotes CA and CDCA production”.

10. The definition of FBG is missing. Besides, in Figure 2, 3 and 5 some graphs (2c-e) actually include 2 graphs. This is not properly explained in the Figures captions. As I commented before, the figures contain too much information, so it is quite difficult to interpret them.

11.     In y opinion the statement would be “…..was partially restored after administration…”

12.     Enzymatic kits for analyzing biochemical parameters are mentioned twice. I suggest removing them from section 4.1 and keeping them in section 4.4.

13.    What internal standard do they add?

14.   How do they analyze biochemical parameters in HepG2 cells?

 Please, go to the revised manuscript to locate what each comment refers to, the phrases and paragraphs that, in my opinion, should go in "Discussion" and other minor comments
